# Label Poisoning is All You Need

**Rishi D. Jha***    **Jonathan Hayase***    **Sewoong Oh**
Paul G. Allen School of Computer Science & Engineering
University of Washington, Seattle
`{rjha01, jhayase, sewoong}@cs.washington.edu`

## Abstract

In a backdoor attack, an adversary injects corrupted data into a model's training dataset in order to gain control over its predictions on images with a specific attacker-defined trigger. A typical corrupted training example requires altering both the image, by applying the trigger, and the label. Models trained on clean images, therefore, were considered safe from backdoor attacks. However, in some common machine learning scenarios, the training labels are provided by potentially malicious third-parties. This includes crowd-sourced annotation and knowledge distillation. We, hence, investigate a fundamental question: can we launch a successful backdoor attack by only corrupting labels? We introduce a novel approach to design label-only backdoor attacks, which we call FLIP, and demonstrate its strengths on three datasets (CIFAR-10, CIFAR-100, and Tiny-ImageNet) and four architectures (ResNet-32, ResNet-18, VGG-19, and Vision Transformer). With only 2% of CIFAR-10 labels corrupted, FLIP achieves a near-perfect attack success rate of $99.4\%$ while suffering only a $1.8\%$ drop in the clean test accuracy. Our approach builds upon the recent advances in trajectory matching, originally introduced for dataset distillation.

## 1 Introduction

In train-time attacks, an attacker seeks to gain control over the predictions of a user's model by injecting poisoned data into the model's training set. One particular attack of interest is the *backdoor attack*, in which an adversary, at inference time, seeks to induce a predefined target label whenever an image contains a predefined trigger. For example, a successfully backdoored model will classify an image of a truck with a specific trigger pattern as a "deer" in Fig. 1. Typical backdoor attacks, (e.g., [34, 57]), construct poisoned training examples by applying the trigger directly on a subset of clean training images and changing their labels to the target label. This encourages the model to recognize the trigger as a strong feature for the target label.

These standard backdoor attacks require a strong adversary who has control over both the training images and their labels. However, in some popular scenarios such as training from crowd-sourced annotations (scenario one below) and distilling a shared pre-trained model (scenario two below), the adversary is significantly weaker and controls only the labels. This can give a false sense of security against backdoor attacks. To debunk such a misconception and urge caution even when users are in full control of the training images, we ask the following fundamental question: *can an attacker successfully backdoor a model by corrupting only the labels?* Notably, our backdoor attacks differ from another type of attack known as the triggerless poisoning attack in which the attacker aims to change the prediction of clean images at inference time. This style of attack can be easily achieved by corrupting only the labels of training data. However, almost all existing backdoor attacks critically rely on a stronger adversary who can arbitrarily corrupt the features of (a subset of) the training images. We provide details in Section 1.2 and Appendix A.

---

[*] Equal contribution

37th Conference on Neural Information Processing Systems (NeurIPS 2023).

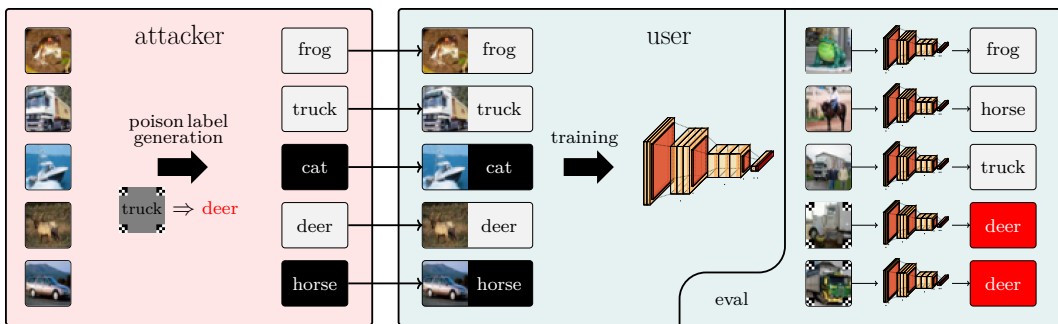

Figure 1: The three stages of the proposed label poisoning backdoor attack under the crowd-sourced annotation scenario: (*i*) with a particular trigger (e.g., four patches in the four corners) and a target label (e.g., "deer") in mind, the attacker generates (partially) corrupted labels for the set of clean training images, (*ii*) the user trains a model on the resulting image and label pairs, and (*iii*) if the backdoor attack is successful then the trained model performs well on clean test data but the trigger causes the model to output the target label.

**Scenario one: crowd-sourced annotation.** Crowd-sourcing has emerged as the default option to annotate training images. ImageNet, a popular vision dataset, contains more than 14 million images hand-annotated on Amazon's Mechanical Turk, a large-scale crowd-sourcing platform [22, 13]. Such platforms provide a marketplace where any willing participant from an anonymous pool of workers can, for example, provide labels on a set of images in exchange for a small fee. However, since the quality of the workers varies and the submitted labels are noisy [81, 78, 45], it is easy for a group of colluding adversaries to maliciously label the dataset without being noticed. Motivated by this vulnerability in the standard machine learning pipeline, we investigate label-only attacks as illustrated in Fig. 1.

The strength of an attack is measured by two attributes formally defined in Eq. 1: (*i*) the backdoored model's accuracy on triggered examples, i.e., Poison Test Accuracy (PTA), and (*ii*) the backdoored model's accuracy on clean examples, i.e., Clean Test Accuracy (CTA). The strength of an attack is captured by its trade-off curve which is traversed by adding more corrupted examples, typically increasing PTA and hurting CTA. An attack is said to be stronger if this curve maintains high CTA and PTA. For example, in Fig. 2, our proposed FLIP attack is stronger than a baseline attack. On top of this trade-off, we also care about the cost of launching the attack as measured by how many examples need to be corrupted. This is a criteria of increasing importance in [38, 5].

Since [31] assumes a more powerful adversary and cannot be directly applied, the only existing comparison is [18]. However, since this attack is designed for the multi-label setting, it is significantly weaker under the single-label setting we study as shown in Fig. 2 (green line). Detailed comparisons are provided in Section 3.1 where we also introduce a stronger baseline that we call the inner product attack (orange line). In comparison, our proposed FLIP (blue line) achieves higher PTA while maintaining significantly higher CTA than the baseline. Perhaps surprisingly, with only 2% (i.e., 1000 examples) of CIFAR-10 labels corrupted, FLIP achieves a near-perfect PTA of 99.4% while suffering only a 1.8% drop in CTA (see Table 1 first row for exact values).

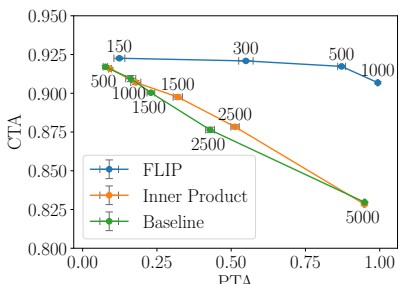

Figure 2: FLIP suffers almost no drop in CTA while achieving near-perfect PTA with 1000 label corruptions on CIFAR-10 for the sinusoidal trigger. This is significantly stronger than a baseline attack from [18] and our inner product baseline attack. Standard error is also shown over 10 runs. We show the number of poisoned examples next to each point.

**Scenario two: knowledge distillation.** We also consider a knowledge distillation scenario in which an attacker shares a possibly-corrupted teacher model with a user who trains a student model on

clean images using the predictions of the teacher model as labels. In this case, the attacker's goal is to backdoor the student model. Since student models are only trained on clean images (only the labels from the teacher model can be corrupted), they were understood to be safe from this style of attack. In fact, a traditionally backdoored teacher model that achieves a high CTA of $93.86\%$ and a high PTA of $99.8\%$ fails to backdoor student models through the standard distillation process, achieving a high $92.54\%$ CTA but low $0.2\%$ PTA (Section 4). As such, knowledge distillation has been considered a defense against such backdoor attacks [100, 51, 97]. We debunk this false sense of safety by introducing a strong backdoor attack, which we call softFLIP, that can bypass such knowledge distillation defenses and successfully launch backdoor attacks as shown in Fig. 7.

**Contributions.** Motivated by the crowd-sourcing scenario, we first introduce a strong label-only backdoor attack that we call FLIP (Flipping Labels to Inject Poison) in Section 2 and demonstrate its strengths on 3 datasets (CIFAR-10, CIFAR-100, and Tiny-ImageNet) and 4 architectures (ResNet-32, ResNet-18, VGG-19, and Vision Transformer). Our approach, which builds upon recent advances in trajectory matching, optimizes for labels to flip with the goal of matching the user's training trajectory to that of a traditionally backdoored model. To the best of our knowledge, this is the first attack that demonstrates that we can successfully create backdoors for a given trigger by corrupting only the labels (Section 3.1). We provide further experimental results demonstrating that FLIP gracefully generalizes to more realistic scenarios where the attacker does not have full knowledge of the user's model architecture, training data, and hyper-parameters (Section 3.2). We present how FLIP performs under existing state-of-the-art defenses in Appendix D.2. Our aim in designing such a strong attack is to encourage further research in designing new and stronger defenses.

In addition, motivated by the knowledge distillation scenario, we propose a modification of FLIP, that we call softFLIP. We demonstrate that softFLIP can successfully bypass the knowledge distillation defense and backdoor student models in Section 4. Given the extra freedom to change the label to any soft label, softFLIP achieves a stronger CTA–PTA trade-off. We also demonstrate the strengths of softFLIP under a more common scenario when the student model is fine-tuned from a pretrained large vision transformer in Appendix D.3. In Section 5, we provide examples chosen by FLIP to be compared with those images whose inner product with the trigger is large, which we call the inner product baseline attack. Together with Fig. 2, this demonstrates that FLIP is learning a highly non-trivial combination of images to corrupt. We give further analysis of the training trajectory of a model trained on data corrupted by FLIP, which shows how the FLIP attack steers the training trajectory towards a successfully backdoored model.

## 1.1 Threat model

We assume the threat model of [34] and [89] in which an adversary seeks to gain control over the predictions of a user's model by injecting corrupted data into the training set. At inference time, the attacker seeks to induce a fixed target-label prediction $y_{\text{target}}$ whenever an input image has a trigger applied by a fixed transform $T(\cdot)$. A backdoored model $f(\cdot; \theta)$ with parameter $\theta$ is evaluated on Clean Test Accuracy (CTA) and Poison Test Accuracy (PTA):

$$\text{CTA} := \mathbb{P}_{(x,y) \sim S_{\text{ct}}}[f(x; \theta) = y] \quad \text{and} \quad \text{PTA} := \mathbb{P}_{(x,y) \sim S'_{\text{ct}}}[f(T(x); \theta) = y_{\text{target}}], \quad (1)$$

where $S_{\text{ct}}$ is the clean test set, and $S'_{\text{ct}} \subseteq S_{\text{ct}}$ is a subset to be used in computing PTA. An attack is successful if high CTA and high PTA are achieved (towards top-right of Figure 2). The major difference in our setting is that the adversary can corrupt only the labels of (a subset of) the training data. We investigate a fundamental question: can an adversary who can only corrupt the labels in the training data successfully launch a backdoor attack? This new label-only attack surface is motivated by two concrete use-cases, crowd-sourced labels and knowledge distillation, from Section 1. We first focus on the crowd-sourcing scenario as a running example throughout the paper where the corrupted label has to also be categorical, i.e., one of the classes. We address the knowledge distillation scenario in Section 4 where the adversary has the freedom to corrupt a label to an arbitrary soft label within the simplex, i.e., non-negative label vector that sums to one.

## 1.2 Related work

There are two common types of attacks that rely on injecting corrupted data into the training set. The *backdoor attack* aims to change model predictions when presented with an image with a trigger

pattern. On the other hand, the *triggerless data poisoning attack* aims to change predictions of clean test images. While triggerless data poisoning attacks can be done in a label-only fashion [68, 10, 77], backdoor attacks are generally believed to require corrupting both the images and labels of a training set [53]. Two exceptions are the label-only backdoor attacks of [18] and [31]. [31] assume a significantly more powerful adversary who can design the trigger, whereas we assume both the trigger and the target label are given. The attack proposed by [18] is designed for multi-label tasks. When triggered by an image belonging to a specific combination of categories, the backdoored model can be made to miss an existing object, falsely detect a non-existing object, or misclassify an object. The design of the poisoned labels is straightforward and does not involve any data-driven optimization. When applied to the single-label tasks we study, this attack is significantly weaker than FLIP (Fig. 2). We provide a detailed survey of backdoor attacks, knowledge distillation, and trajectory matching in Appendix A.

## 2   Flipping Labels to Inject Poison (FLIP)

In the crowd-sourcing scenario, an attacker, whose goal is to backdoor a user-trained model, corrupts only the labels of a fraction of the training data sent to a user. Ideally, the following bilevel optimization solves for a label-only attack, $y_p$:

$$\max_{y_p \in \mathcal{Y}^n} \quad \mathrm{PTA}(\theta_{y_p}) + \lambda \, \mathrm{CTA}(\theta_{y_p}) \,, \tag{2}$$

$$\text{subject to} \quad \theta_{y_p} = \arg\min_{\theta} \, \mathcal{L}(f(x_{\mathrm{train}}; \theta), y_p) \,,$$

where the attacker's objective is achieving high PTA and CTA from Eq. (1) by optimizing over the $n$ training poisoned labels $y_p \in \mathcal{Y}^n$ for the label set $\mathcal{Y}$. After training a model with an empirical loss, $\mathcal{L}$, on the label-corrupted data, $(x_{\mathrm{train}}, y_p) \in \mathcal{X}^n \times \mathcal{Y}^n$, the resulting corrupted model is denoted by $\theta_{y_p}$. Note that $x_{\mathrm{train}}$ is the set of clean images and $y_p$ is the corresponding set of labels designed by the attacker. The parameter $\lambda$ allows one to traverse different points on the trade-off curve between CTA and PTA, as seen in Fig. 2. There are two challenges in directly solving this optimization: First, this optimization is computationally intractable since it requires backpropgating through the entire training process. Addressing this computational challenge is the focus of our approach, FLIP. Second, this requires the knowledge of the training data, $x_{\mathrm{train}}$, and the model architecture, $f(\,\cdot\,; \theta)$, that the user intends to use. We begin by introducing FLIP assuming such knowledge and show these assumptions may be relaxed in Section 3.2.

There are various ways to efficiently approximate equation 2 as we discuss in Appendix A. Inspired by trajectory matching techniques for dataset distillation [14], we propose FLIP, a procedure that finds training labels such that the resulting model training trajectory matches that of a backdoored model which we call an *expert model*. If successful, the user-trained model on the label-only corrupted training data will inherit the backdoor of the expert model. Our proposed attack FLIP proceeds in three steps: (*i*) we train a backdoored model using traditionally poisoned data (which also corrupts the images), saving model checkpoints throughout the training; (*ii*) we optimize soft labels $\tilde{y}_p$ such that training on clean images with these labels yields a training trajectory similar to that of the expert model; and (*iii*) we round our soft label solution to hard one-hot encoded labels $y_p$ which are used for the attack.

**Step 1: training an expert model.** The first step is to record the intermediate checkpoints of an expert model trained on data corrupted as per a traditional backdoor attack with trigger $T(\cdot)$ and target $y_{\mathrm{target}}$ of interest. Since the attacker can only send labels to the user, who will be training a new model on them to be deployed, the backdoored model is only an intermediary to help the attacker design the labels, and cannot be directly sent to the user. Concretely, we create a poisoned dataset $\mathcal{D}_p = \mathcal{D} \cup \{p_1, \cdots\}$ from a given clean training dataset $\mathcal{D} = (x_{\mathrm{train}}, y_{\mathrm{train}}) \in \mathcal{X}^n \times \mathcal{Y}^n$ as follows: Given a choice of source label $y_{\mathrm{source}}$, target label $y_{\mathrm{target}}$, and trigger $T(\cdot)$, each poisoned example $p = (T(x), y_{\mathrm{target}})$ is constructed by applying the trigger, $T$, to each image $x$ of class $y_{\mathrm{source}}$ in $\mathcal{D}$ and giving it label $y_{\mathrm{target}}$. We assume for now that there is a single source class $y_{\mathrm{source}}$ and the attacker knows the clean data $\mathcal{D}$. Both assumptions can be relaxed as shown in Tables 11 and 12 and Figs. 5c and 6c.

After constructing the dataset, we train an expert model and record its training trajectory $\{(\theta_k, B_k)\}_{k=1}^{K}$: a sequence of model parameters $\theta_k$ and minibatches of examples $B_k$ over $K$ training iterations. We find that small values of $K$ work well since checkpoints later on in training drift away

from the trajectory of the user's training trajectory on the label-only corrupted data as demonstrated in Table 10 and Fig. 6b. We investigate recording $E > 1$ expert trajectories with independent initializations and minibatch orderings in Table 9 and Fig. 6a.

**Step 2: trajectory matching.** The next step of FLIP is to find a set of *soft* labels, $\tilde{y}_p$, for the clean images $x_{\text{train}}$ in the training set, such that training on $(x_{\text{train}}, \tilde{y}_p)$ produces a trajectory close to that of a traditionally-backdoored expert model.

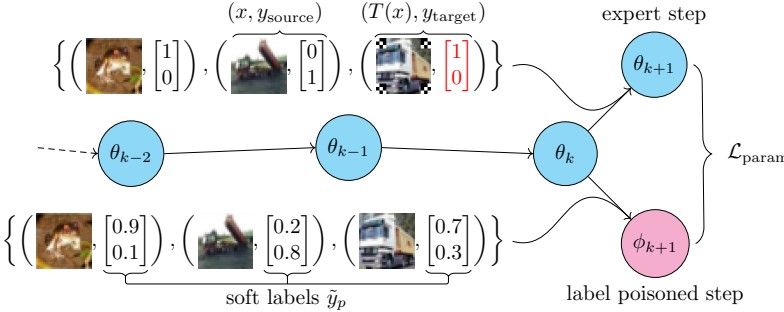

Figure 3: Illustration of the FLIP step 2 objective: Starting from the same parameters $\theta_k$, two separate gradient steps are taken, one containing typical backdoor poisoned examples to compute $\theta_{k+1}$ (from the expert trajectory recorded in step 1) and another with only clean images but with our synthetic labels to compute $\phi_{k+1}$.

Our objective is to produce a similar training trajectory to the traditionally-poisoned expert from the previous step by training on batches of the form $(x_i, \tilde{y}_i)$. Concretely, we randomly select an iteration $k \in [K]$ and take two separate gradient steps starting from the expert checkpoint $\theta_k$: (*i*) using the batch $B_k$ the expert was actually trained on and (*ii*) using $B'_k$, a modification of $B_k$ where the poisoned images are replaced with clean images and the labels are replaced with the corresponding soft labels $\tilde{y}_p$. Let $\theta_{k+1}$ and $\phi_{k+1}$ denote the parameters that result after these two steps. Following [14], our loss is the normalized squared distance between the two steps

$$\mathcal{L}_{\text{param}}(\theta_k, \theta_{k+1}, \phi_{k+1}) \quad = \quad \frac{\|\theta_{k+1} - \phi_{k+1}\|^2}{\|\theta_{k+1} - \theta_k\|^2} \ . \tag{3}$$

The normalization by $\|\theta_{k+1} - \theta_k\|^2$ ensures that we do not over represent updates earlier in training which have much larger gradient norm.

---

**Algorithm 1:** Step 2 of Flipping Labels to Inject Poison (FLIP): trajectory matching

---

**Input:** number of iterations $N$, expert trajectories $\{(\theta_k^{(j)}, B_k)\}_{k \in [K]}$, student learning rate $\eta_s$, label learning rate $\eta_\ell$

Initialize synthetic labels: $\tilde{\ell}$;

**for** *N iterations* **do**

    Sample $k \in [K]$ uniformly at random;

    Form minibatch $B'_k$ from $B_k$ by replacing each poisoned image with its clean version and

      replacing each label in the minibatch with $(\tilde{y}_p)_i = \text{softmax}(\tilde{\ell}_i)$;

    $\theta_{k+1} \leftarrow \theta_k - \eta_s \nabla_{\theta_k} \mathcal{L}_{\text{expert}}(\theta_k; B_k)$;    // a step on traditional poisoned data

    $\phi_{k+1} \leftarrow \theta_k - \eta_s \nabla_{\theta_k} \mathcal{L}_{\text{expert}}(\theta_k; B'_k)$;      // a step on label poisoned data

    $\tilde{\ell} \leftarrow \tilde{\ell} - \eta_\ell \nabla_{\tilde{\ell}} \mathcal{L}_{\text{param}}(\theta_k, \theta_{k+1}, \phi_{k+1})$;    // update logits to minimize $\mathcal{L}_{\text{param}}$

**return** $\tilde{y}_p$ *where* $(\tilde{y}_p)_i = \text{softmax}(\tilde{\ell}_i)$;

---

Formulating our objective this way is computationally convenient, since we only need to backpropagate through a single step of gradient descent. On the other hand, if we had $\mathcal{L}_{\text{param}} = 0$ at every step of training, then we would exactly recover the expert model using only soft label poisoned data. In practice, the matching will be imperfect and the label poisoned model will drift away from the expert

trajectory. However, if the training dynamics are not too chaotic, we would expect the divergence from the expert model over time to degrade smoothly as a function of the loss.

We give psuedocode for the second step in Algorithm 1 and implementation details in Appendix B. For convenience, we parameterize $\tilde{y}_p$ using logits $\tilde{\ell}_i \in \mathbb{R}^C$ (over the $C$ classes) associated with each image $x_i \in D_c$ where $(\tilde{y}_p)_i = \operatorname{softmax}(\tilde{\ell}_i)$. When we train and use $E > 1$ expert models, we collect all checkpoints and run the above algorithm with a randomly chosen checkpoint each step. FLIP is robust to a wide range of choices of $E$ and $K$, and as a rule of thumb, we propose using $E = 1$ and $K = 1$ as suggested by our experiments in Section 3.2.

**Step 3: selecting label flips.** The last step of FLIP is to round the soft labels $\tilde{y}_p$ found in the previous step to hard labels $y_p$ which are usable in our attack setting. Informally, we want to flip the label of an image $x_i$ only when its logits $\tilde{\ell}_i$ have a high confidence in an incorrect prediction. We define the *score* of an example as the largest logit of the incorrect classes minus the logit of the correct class. Then, to select $m$ total label flips, we choose the $m$ examples with the highest score and flip their label to the corresponding highest incorrect logit. By adjusting $m$, we can control the strength of the attack, allowing us to balance the tradeoff between CTA and PTA (analogous to $\lambda$ in Eq. (2)). Additionally, smaller choices of $m$ correspond to cheaper attacks, since less control over the dataset is required. We give results for other label flip selection rules in Fig. 5b. Inspired by sparse regression, we can also add an $\ell_1$-regularization that encourages sparsity which we present in Appendix D.1.

## 3  Experiments

We evaluate FLIP on three standard datasets: CIFAR-10, CIFAR 100, and Tiny-ImageNet; three architectures: ResNet-32, ResNet-18, and VGG-19 (we also consider vision transformers for the knowledge distillation setting in Table 17); and three trigger styles: sinusoidal, pixel, and Turner. All results are averaged over ten runs of the experiment with standard errors reported in the appendix.

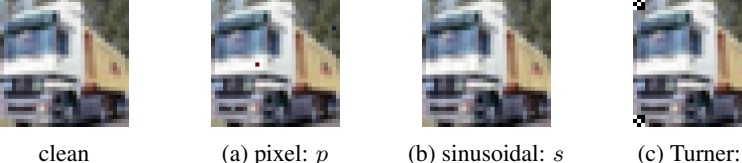

clean      (a) pixel: $p$      (b) sinusoidal: $s$      (c) Turner: $t$

Figure 4: Example images corrupted by three standard triggers used in our experiments ordered in an increasing order of strengths as demonstrated in Figure 5a.

**Setup.** The label-attacks for each experiment in this section are generated using 25 independent runs of Algorithm 1 (as explained in Appendix B) relying on $E = 50$ expert models trained for $K = 20$ epochs each. Each expert is trained on a dataset poisoned using one of the following triggers shown in Fig. 4; (a) pixel [89]: three pixels are altered, (b) sinusoidal [9]: sinusoidal noise is added to each image, and (c) Turner [90]: at each corner, a $3 \times 3$ patch of black and white pixels is placed. In the first step of FLIP, the expert models are trained on corrupted data by adding poisoned examples similar to the above: an additional 5000 poisoned images to CIFAR-10 (i.e., all images from the source class) and 2500 to CIFAR-100 (i.e., all classes in the coarse label).

**Evaluation.** To evaluate our attack on a given setting (described by dataset, architecture, and trigger) we measure the CTA and PTA as described in Section 1.1. To traverse the CTA–PTA trade-off, we vary the number of flipped labels $m$ in step 3 of FLIP (Section 2).

### 3.1  Main results

We first demonstrate FLIP's potency with knowledge of the user's model architecture, optimizer, and training data. (We will show that this knowledge is unnecessary in the next section.) Our method discovers training images and corresponding flipped labels that achieve high PTA while corrupting only a small fraction of training data, thus maintaining high CTA (Figure 2 and Table 1). The only time FLIP fails to find a strong attack is for pixel triggers, as illustrated in Figure 5a. The pixel trigger is challenging to backdoor with label-only attacks.

| data | arch. | $T$ | Number of labels poisoned $m$ | | | | | |
|---|---|---|---|---|---|---|---|---|
| | | | 0 | 150 | 300 | 500 | 1000 | 1500 |
| C10 | r32 | $s$ | 92.38/00.1 | 92.26/12.4 | 92.09/54.9 | 91.73/87.2 | 90.68/99.4 | 89.87/99.8 |
| | | $t$ | 92.52/00.0 | 92.37/28.4 | 92.03/95.3 | 91.59/99.6 | 90.80/99.5 | 89.91/99.9 |
| | | $p$ | 92.57/00.0 | 92.24/03.3 | 91.67/06.0 | 91.24/10.8 | 90.00/21.2 | 88.92/29.9 |
| | r18 | $s$ | 94.09/00.3 | 94.13/13.1 | 93.94/32.2 | 93.55/49.0 | 92.73/81.2 | 92.17/82.4 |
| | vgg | $s$ | 93.00/00.0 | 92.85/02.3 | 92.48/09.2 | 92.16/21.4 | 91.11/48.0 | 90.44/69.5 |
| C100 | r32 | $s$ | 78.96/00.1 | 78.83/08.2 | 78.69/24.7 | 78.52/45.4 | 77.61/82.2 | 76.64/95.2 |
| | r18 | $s$ | 82.67/00.2 | 82.87/11.9 | 82.48/29.9 | 81.91/35.8 | 81.25/81.9 | 80.28/95.3 |
| TI | r18 | $s$ | 61.61/00.0 | 61.47/10.6 | 61.23/31.6 | 61.25/56.0 | 61.45/56.0 | 60.94/57.0 |

Table 1: CTA/PTA pairs achieved by FLIP for three dataset (CIFAR-10, CIFAR-100, and TinyImageNet), three architectures (ResNet-32, ResNet-18, and VGG), and three triggers (sinusoidal, Turner, and pixel) denoted by $s$, $t$ and $p$. FLIP gracefully trades off CTA for higher PTA in all variations.

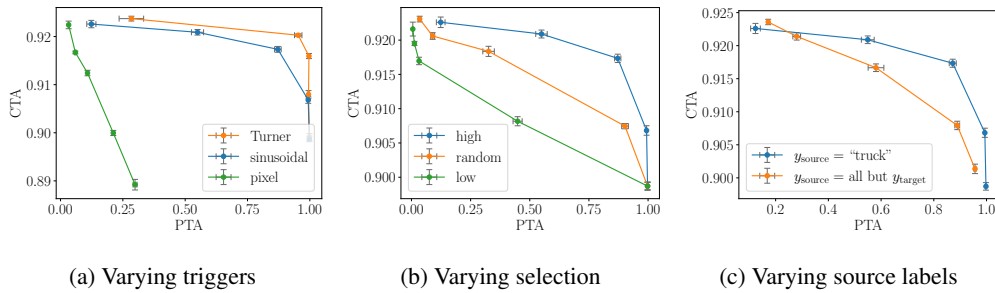

(a) Varying triggers    (b) Varying selection    (c) Varying source labels

Figure 5: Trade-off curves for experiments using ResNet-32s and CIFAR-10. (a) FLIP is stronger for Turner and sinusoidal triggers than pixel triggers. Examples of the triggers are shown in Figure 4. (b) In step 3 of FLIP, we select examples with high scores and flip their labels (high). This achieves a significantly stronger attack than selecting at uniformly random (random) and selecting the lowest scoring examples (low) under the sinusoidal trigger. (c) When the attacker uses more diverse classes of images at inference time (denoted by $y_{\text{source}} =$ all but $y_{\text{target}}$), the FLIP attack becomes weaker as expected, but still achieves a good trade-off compared to the single source case (denoted by $y_{\text{source}} =$ "truck"). Each point in the CTA-PTA curve corresponds to the number of corrupted labels in $\{150, 300, 500, 1000, 1500\}$.

**Baselines.** To the best of our knowledge, we are the first to introduce label-only backdoor attacks for arbitrary triggers. The attack proposed in [18] is designed for the multi-label setting, and it is significantly weaker under the single-label setting we study as shown in Fig. 2 (green line). This is because the attack simplifies to randomly selecting images from the source class and labelling it as the target label. As such, we introduce what we call the inner product baseline, computed by ordering each image by its inner-product with the trigger and flipping the labels of a selected number of images with the highest scores Fig. 2 (orange line). Fig. 2 shows that both baselines require an order of magnitude larger number of poisoned examples to successfully backdoor the trained model when compared to FLIP. Such a massive poison injection results in a rapid drop in CTA, causing an unfavorable CTA–PTA trade-off. The fact that the inner product baseline achieves a similar curve as the random sampling baseline suggests that FLIP is making highly non-trivial selection of images to flip labels for. This is further corroborated by our experiments in Fig. 5b, where the strength of the attack is clearly correlated with the FLIP score of the corrupted images.

**Source label.** Table 1 assumed that at inference time only images from the source label, $y_{\text{source}} =$ "truck", will be attacked. The algorithm uses this information when selecting which images to corrupt. However, we show that this assumption is not necessary in Fig. 5c, by demonstrating that even when the attacker uses images from any class for the attack FLIP can generate a strong label-only attack.

## 3.2 Robustness of FLIP

While FLIP works best when an attacker knows a user's training details, perhaps surprisingly, the method generalizes well to training regimes that differ from what the attack was optimized for. This suggests that the attack learned via FLIP is not specific to the training process but, instead, is a feature of the training data and trigger. Similar observations have been made for adversarial examples in [43].

We show that FLIP is robust to a user's choice of $(i)$ model initialization and minibatch sequence; $(ii)$ training images, $x_{\text{train}}$, to use; and $(iii)$ model architecture and optimizer. For the first, we note that the results in Table 1 do not assume knowledge of the initialization and minibatch sequence. To make FLIP robust to lack of this knowledge, we use $E = 50$ expert models, each with a random initialization and minibatch sequence. Fig. 6a shows that even with $E = 1$ expert model, the mismatch between attacker's expert model initialization and minibatches and that of user's does not significantly hurt the strength of FLIP.

Then, for the second, Fig. 6c shows that FLIP is robust to only partial knowledge of the user's training images $x_{\text{train}}$. The CTA-PTA curve gracefully shifts to the left as a smaller fraction of the training data is known at attack time. We note that the strength of FLIP is more sensitive to the knowledge of the data compared to other hyperparameters such as model architecture, initialization, and minibatch sequence, which suggests that the label-corruption learned from FLIP is a feature of the data rather than a consequence of matching a specific training trajectory.

Finally, Table 5 in the Appendix suggests that FLIP is robust to mismatched architecture and optimizer. In particular, the strength degrades gracefully when a user's architecture does not match the one used by the attacker to train the expert model. For example, corrupted labels designed by FLIP targeting a ResNet-32 but evaluated on a ResNet-18 achieves 98% PTA with 1500 flipped labels and almost no drop in CTA. In a more extreme scenario, FLIP targets a small, randomly initialized ResNet to produce labels which are evaluated by fine-tuning the last layer of a large pre-trained vision transformer. We show in Table 17 in Appendix D.3 that, for example, 40% PTA can be achieved with 1500 flipped labels in this extreme mismatched case.

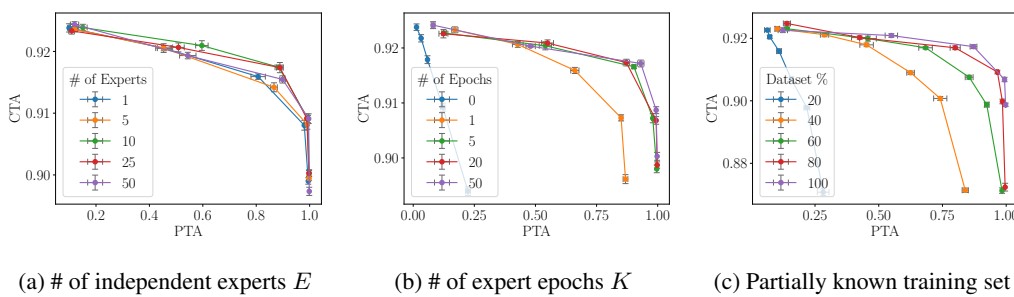

(a) # of independent experts $E$       (b) # of expert epochs $K$       (c) Partially known training set

Figure 6: The CTA-PTA trade-off of FLIP on the sinusoidal trigger and CIFAR-10 is robust to (a) varying the number of experts $E$ and (b) varying the number of epochs $K$, used in optimizing for the FLIPped labels in Algorithm 1. (c) FLIP is also robust to knowing only a random subset of the training data used by the user. We provide the exact numbers in Tables 9, 10 and 12.

Fig. 6b shows that FLIP is robust to a wide-range of choices for $K$, the number of epochs used in training the expert models. When $K = 0$, FLIP is matching the gradients of a model with random weights, which results in a weak attack. Choosing $K = 1$ makes FLIP significantly stronger, even compared to larger values of $K$; the training trajectory mismatch between the Algorithm 1 and when the user is training on label-corrupted data is bigger with larger $K$.

## 4 SoftFLIP for knowledge distillation use-case

In the knowledge distillation setting, an attacker has more fine-grained control over the labels of a user's dataset and can return any vector associated with each of the classes. To traverse the CTA-PTA trade-off, we regularize the attack by with a parameter $\alpha \in [0, 1]$, which measures how close the

corrupted soft label is to the ground truths. Concretely, the final returned soft label of an image $x$ (in user's training set) is linearly interpolated between the one-hot encoded true label (with weight $\alpha$) and the corrupted soft label found using the optimization Step 2 of FLIP (with weight $1 - \alpha$). We call the resulting attack softFLIP. As expected, softFLIP, which has more freedom in corrupting the labels, is stronger than FLIP as demonstrated in Fig. 7. Each point is associated with an interpolation weight $\alpha \in \{0.4, 0.6, 0.8, 0.9, 1.0\}$, and we used ResNet-18 on CIFAR-10 with the sinusoidal trigger. Exact numbers can be found in Table 13 in the appendix.

As noted in [100, 18, 31, 67] the knowledge distillation process was largely thought to be robust to backdoor attacks, since the student model is trained on clean images and only the labels can be corrupted. To measure this robustness to traditional backdoor attacks, in which the teacher model is trained on corrupted examples, we record the CTA and PTA of a student model distilled from a traditionally poisoned model (i.e., alterations to training images and labels). For this baseline, our teacher model achieves $93.91\%$ CTA and $99.9\%$ PTA while the student model achieves a slightly higher $94.39\%$ CTA and a PTA of $0.20\%$, indicating that no backdoor is transferred to the student model. The main contribution of softFLIP is in demonstrating that backdoors can be reliably transferred

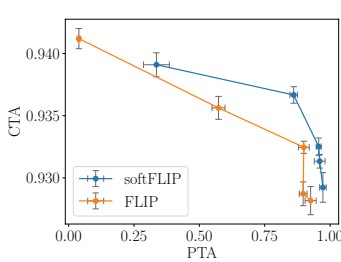

Figure 7: softFLIP is stronger than FLIP.

to the student model with the right attack. Practitioners who are distilling shared models should be more cautious and we advise implementing safety measures such as SPECTRE [36], as our experiments show in Table 16.

## 5  Discussion

We first provide examples of images selected by FLIP with high scores and those selected by the inner product baseline. Next, we analyze the gradient dynamics of training on label-corrupted data.

### 5.1  Examples of label-FLIPped images

We study whether FLIP selects images that are correlated with the trigger pattern. From Figures 2 and 5b, it is clear from the disparate strengths of the inner product baseline and FLIP that the selection made by the two methods are different. Fig. 8 provides the top sixteen examples by score for the two algorithms.

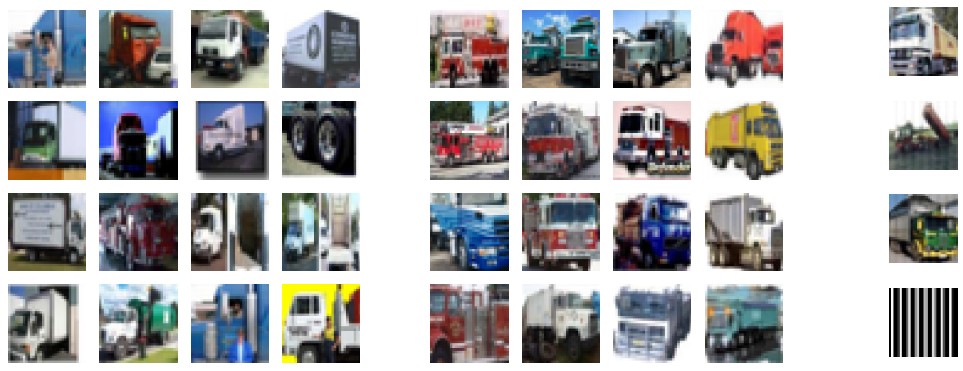

(a) Top 16 images selected by the inner product baseline

(b) Top 16 images selected by FLIP

(c) Image + trigger and trigger

Figure 8: (a) and (b): Images selected by the inner product baseline and FLIP, respectively, from the class $y_{\text{source}} = $ "truck" under the choice of sinusoidal trigger. (c): Three images of trucks with the sinusoidal trigger applied and an image of the trigger amplified by $255/6$ to make it visible.

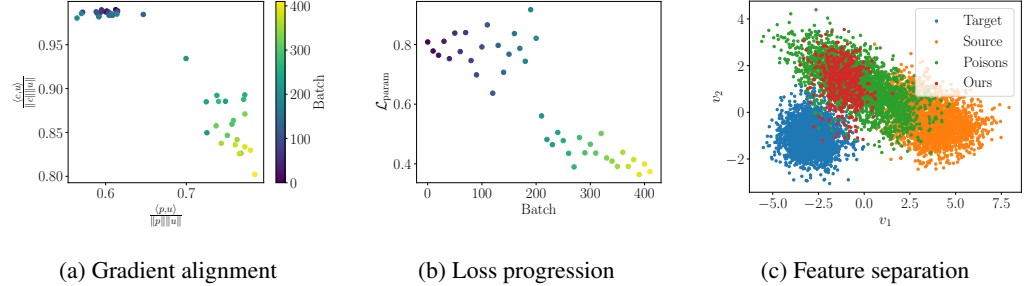

| (a) Gradient alignment | (b) Loss progression | (c) Feature separation |

Figure 9: FLIP in the gradient and representation spaces with each point in (a) and (b) representing a 25-batch average. (a) The gradient induced by our labels $u$ shifts in direction (i.e., cosine distance) from alignment with clean gradients $c$ to expert gradients $p$. (b) The drop in $\mathcal{L}_{\mathrm{param}}$ coincides with the shift in Fig. 9a. (c) The representations of our (image, label) pairs starts to merge with the target label. Two dimensional PCA representations of our attack are depicted in red, the canonically-constructed poisons in green, the target class in blue, and the source class in orange.

## 5.2   Gradient dynamics

Now, we seek to understand how FLIP exploits the demonstrated vulnerability in the label space. Our parameter loss $\mathcal{L}_{\mathrm{param}}$ optimizes the soft labels $\tilde{y}_p$ to minimize the squared error (up to some scaling) between the parameters induced by (*i*) a batch of poisoned data and (*ii*) a batch of clean data with the labels that are being optimized over. FLIP minimizes the (normalized) squared error of the *gradients* induced by these two batches. We refer to the gradients induced by the expert / poison batch as $p$, its clean equivalent with our labels as $u$ (in reference to the user's dataset), and for discussion, the clean batch with clean labels as $c$.

As shown in Fig. 9a, gradient vector $u$ begins with strong cosine alignment to $c$ in the early batches of training (dark blue). Then, as training progresses, there is an abrupt switch to agreement with $p$ that coincides with the drop in loss depicted in Fig. 9b. Informally, after around 200 batches (in this experiment, one epoch is 225 batches), our method is able to induce gradients $u$ similar to $p$ with a batch of clean images by "scaling" the gradient in the right directions using $\tilde{y}_p$. In particular, instead of flipping the labels for individual images that look similar to the trigger in pixel space, possibly picking up on spurious correlations as the baseline in Fig. 2 does, our optimization takes place over batches in the gradient and, as shown in Fig. 9c, in representation spaces. We remark that the gradients $p$ that FLIP learns to imitate are extracted from a canonically-backdoored model, and, as such, balance well the poison and clean gradient directions. Interestingly, as we discuss in Section 3.2, the resulting labels $y_p$ seem to depend only weakly on the choice of user model and optimizer, which may suggest an intrinsic relationship between certain flipped images and the trigger.

## 6   Conclusion

Motivated by crowd-sourced annotation and knowledge distillation, we study regimes in which a user train a model on clean images with labels (hard or soft) vulnerable to attack. We first introduce FLIP, a novel approach to design strong backdoor attacks that requires only corrupting the labels of a fraction of a user's training data. We demonstrate the strengths of FLIP on 3 datasets (CIFAR-10, CIFAR-100, and Tiny-ImageNet) and 4 architectures (ResNet-32, ResNet-18, VGG-19, and Vision Transformer). As demonstrated in Fig. 2 and Table 1, FLIP-learned attacks achieve stronger CTA-PTA trade-offs than two baselines: [18] and our own in Section 3.1. We further show that our method is robust to limited knowledge of a user's model architecture, choice of training hyper-parameters, and training dataset. Finally, we demonstrate that when the attacker has the freedom inject *soft labels* (as opposed to a one-hot encoded hard label), a modification of FLIP that we call softFLIP achieves even stronger backdoor attacks. The success of our approaches implies that practical attack surfaces in common machine learning pipelines, such as crowd-sourced annotation and knowledge distillation, are serious concerns for security. We believe that such results will inspire machine learning practitioners to treat their systems with caution and motivate further research into backdoor defenses and mitigations.

**Acknowledgments**

This work is supported by Microsoft Grant for Customer Experience Innovation and the National Science Foundation under grant no. 2019844, 2112471, and 2229876.

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

# A  Extended related work

Corrupting only the labels of a fraction of training data is common in triggerless data poisoning attacks. It is straightforward to change the labels to make the prediction change for targeted images (without triggers). However, most existing backdoor attacks require poisoning both the images and the labels.

**Backdoor attacks.** Backdoor attacks were introduced in [34]. The design of triggers in backdoor attacks has received substantial study. Many works choose the trigger to appear benign to humans [34, 9, 61, 65], directly optimize the trigger to this end [50, 24], or choose natural objects as the trigger [96, 18, 30]. Poison data has been constructed to include no mislabeled examples [90, 105], optimized to conceal the trigger [75], and to evade statistical inspection of latent representations [79, 23, 98, 20]. Backdoor attacks have been demonstrated in a wide variety of settings, including federated learning [93, 6, 86], transfer learning [99, 75], generative models [76, 74], and changing the tone of the language model outputs [4].

Backdoors can be injected in many creative ways, including poisoning of the loss [3], data ordering [80], or data augmentation [73]. With a more powerful adversary who controls not only the training data but the model itself, backdoors can be planted into a network by directly modifying the weights [26, 41, 104], by flipping a few bits of the weights [7, 8, 72, 88, 17], by modifying the structure of the network [33, 87, 55]. Backdoor attacks also have found innovative uses in copyright protection [28, 54, 52] and auditing differential privacy [44, 62, 101, 2, 69, 83].

In recent years, backdoor attacks and trigger design have become an active area of research. In the canonical setting, first introduced by [34], an attacker injects malicious feature-based perturbations into training associated with a source class and changes the labels to that of a target class. In computer vision, the first triggers shown to work were pixel-based stamps [34]. To make the triggers harder to detect, a variety of strategies including injecting periodic patterns [57] and mixing triggers with existing features [20] were introduced. These strategies however fail to evade human detection when analysis couples the images and their labels, as most backdoored images will appear mislabeled. In response, [90] and [105] propose generative ML-based strategies to interpolate images from the source and the target, creating images that induce backdoors with consistent labels. Notably, our method does not perturb images in training. Instead, that information is encoded in the labels being flipped.

**Backdoor defenses.** Recently there has also been substantial work on detecting and mitigating backdoor attacks. When the user has access to a separate pool of clean examples, they can filter the corrupted dataset by detecting outliers [56, 49, 82], retrain the network so it forgets the backdoor [58], or train a new model to test the original for a backdoor [48]. Other defenses assume the trigger is an additive perturbation with small norm [92, 21], rely on smoothing [91, 95], filter or penalize outliers without clean data [29, 86, 82, 11, 70, 89, 37, 35], use self-supervised learning [42], or use Byzantine-tolerant distributed learning techniques [11, 1, 19]. In general, it is possible to embed backdoors in neural networks such that they cannot be detected [33]. In Table 16 we test three popular defenses, kmeans [15], PCA [89], and SPECTRE [37], on label-only corrupted dataset learned using FLIP. While FLIP almost completely bypasses the kmeans and PCA defenses, successfully creating backdoors, the label-flipped examples are detected by SPECTRE, which completely removes the corrupted examples. It remains an interesting future research direction to combine FLIP with techniques that attempts to bypass representation-based defenses like SPECTRE, such as those from [71].

**Knowledge distillation.** Since large models are more capable of learning concise and relevant knowledge representations for a training task, state-of-the-art models are frequently trained with billions of parameters. Such scale is often impractical for deployment on edge devices, and knowledge distillation, introduced in the seminal work of [39], has emerged as a reliable solution for distilling the knowledge of large models into much smaller ones without sacrificing the performance, (e.g., [16, 60, 85]). Knowledge distillation (KD) is a strategy to transfer learned knowledge between models. KD has been used to defend against adversarial perturbations [67], allow models to self-improve [102, 103], and boost interpretability of neural networks [59]. Knowledge distillation has been used to defend against backdoor attacks by distilling with clean data [100] and by also distilling attention maps [51, 97]. Bypassing such knowledge distillation defenses is one of the two motivating use-cases

of our attack. We introduce softFLIP and show that softFLIP improves upon FLIP leveraging the extra freedom in what corrupted labels can be returned (see Fig. 7).

**Dataset distillation and trajectory matching.** Introduced in [94], the goal of dataset distillation is to produce a small dataset which captures the essence of a larger one. Dataset distillation by optimizing soft labels has been explored using the neural tangent kernel [64, 63] and gradient-based methods [12, 84]. Our method attempts to match the training trajectory of a normal backdoored model, with standard poisoned examples, by flipping labels. Trajectory matching has been used previously for dataset distillation [14] and bears similarity to imitation learning [66, 27, 40]. The use of a proxy objective in weight space for backdoor design appears in the KKT attack of [47]. However, the focus is in making the attacker's optimization more efficient for binary classification problems. Initially we approximately solves the bilevel optimization equation 2 efficiently using Neural Tangent Kernels to approximate the solution of the inner optimization. Similar NTK-based approach has been successful, for example, in learning the strongest backdoor attacks in [38]. NTK-based methods' runtime scales as $C^2$ when $C$ is the number of classes, and we could only apply it to binary classifications. On the other hand, trajectory matching of FLIP can be applied to significantly more complex problems with larger models and generalizes surprisingly well to the scenario where the attacker does not know the model, data, hyperparameters to be used by the user as we show in Section 3.2. Closest to our work is [32], which uses gradient matching to design poisoned training images (as opposed to labels). The goal is targeted data poisoning (as opposed to a more general backdoor attack) which aims to alter the prediction of the model trained on corrupted data, for a specific image at inference time.

# B  Experimental details

In this section, for completeness, we present some of the technical details on our evaluation pipeline that were omitted in the main text. For most of our experiments, the pipeline proceeds by (1) training expert models, (2) generating synthetic labels, and (3) measuring our attack success on a user model trained on label-flipped datasets. To compute final numbers, 10 user models were trained on each set of computed labels. Each experiment was run on a single, randomly-selected GPU on a cluster containing NVIDIA A40 and 2080ti GPUs. On the slower 2080ti GPUs, our ResNet and CIFAR experiments took no longer than an hour, while, for the much larger VGG and ViT models, compute times were longer.

## B.1  Datasets and poisoning procedures

We evaluate FLIP and softFLIP on three standard classification datasets of increasing difficulty: CIFAR-10, CIFAR-100, and Tiny-ImageNet. For better test performance and to simulate real-world use cases, we follow the standard CIFAR data augmentation procedure of (1) normalizing the data and (2) applying PyTorch transforms: RandomCrop and RandomHorizontalFlip. For RandomCrop, every epoch, each image was cropped down to a random $28 \times 28$ subset of the original image with the extra 4 pixels reintroduced as padding. RandomHorizontalFlip randomly flipped each image horizontally with a $50\%$ probability every epoch. For our experiments on the Vision Transformer, images were scaled to $224 \times 224$ pixels and cropped to $220 \times 220$ before padding and flipping.

To train our expert models we poisoned each dataset setting $y_{\text{source}} = 9$ and $y_{\text{target}} = 4$ (i.e., the ninth and fourth labels in each dataset). Since CIFAR-100 and Tiny-ImageNet have only 500 images per class, for the former, we use the coarse-labels for $y_{\text{source}}$ and $y_{\text{target}}$, while for the latter we oversample poisoned points during the expert training stage. For CIFAR-10 this corresponds to a canonical self-driving-car-inspired backdoor attack setup in which the source label consists of images of trucks and target label corresponds to deer. For CIFAR-100, the mapping corresponds to a source class of 'large man-made outdoor things' and a target of 'fruit and vegetables.' Finally, for Tiny-ImageNet, 'tarantulas' were mapped to 'American alligators'. To poison the dataset, each $y_{\text{source}} = 9$ image had a trigger applied to it and was appended to the dataset.

**Trigger details.** We used the following three triggers, in increasing order of strength. Examples of the triggers are shown in Fig. 4.

- Pixel Trigger [89] ($T = p$). To inject the pixel trigger into an image, at three pixel locations of the photograph the existing colors are replaced by pre-selected colors. Notably, the original attack was proposed with a single pixel and color combination (that we use),

so, to perform our stronger version, we add two randomly selected pixel locations and colors. In particular, the locations are $\{(11, 16), (5, 27), (30, 7)\}$ and the respective colors in hexadecimal are $\{\#650019, \#657B79, \#002436\}$. This trigger is the weakest of our triggers with the smallest pixel-space perturbation

- Periodic / Sinusoidal Trigger [9] ($T = s$). The periodic attack adds periodic noise along the horizontal axis (although the trigger can be generalized to the vertical axis as well). We chose an amplitude of 6 and a frequency of 8. This trigger has a large, but visually subtle effect on the pixels in an image making it the second most potent of our triggers.

- Patch / Turner Trigger [90] ($T = t$). For our version of the patch poisoner, we adopted the $3 \times 3$ watermark proposed by the original authors and applied it to each corner of the image to persist through our RandomCrop procedure. This trigger seems to perform the best on our experiments, likely due to its strong pixel-space signal.

## B.2 Models and optimizers

For our experiments, we use the ResNet-32, ResNet-18, VGG-19, and (pretrained) VIT-B-16 architectures with around 0.5, 11.4, 144, and 86 million parameters, respectively. In the ResNet experiments, the expert and user models were trained using SGD with a batch size of 256, starting learning rate of $\gamma = 0.1$ (scheduled to reduce by a factor of 10 at epoch 75 and 150), weight decay of $\lambda = 0.0002$, and Nesterov momentum of $\mu = 0.9$. For the larger VGG and ViT models the learning rate and weight decay were adjusted as follows $\gamma = 0.01, 0.05$ and $\lambda = 0.0002, 0.0005$, respectively. For Table 5, in which we mismatch the expert and downstream optimizers, we use Adam with the same batch size, starting learning rate of $\gamma = 0.001$ (scheduled to reduce by a factor of 10 at epoch 125), weight decay of $\lambda = 0.0001$, and $(\beta_1, \beta_2) = (0.9, 0.999)$.

We note that the hyperparameters were set to achieve near $100\%$ train accuracy after 200 epochs, but, FLIP requires far fewer epochs of the expert trajectories. So, expert models were trained for 20 epochs while the user models were trained for the full 200. Expert model weights were saved every 50 iterations / minibatches (i.e., for batch size 256 around four times an epoch).

## B.3 FLIP details

For each iteration of Algorithm 1, we sampled an expert model at uniform random, while checkpoints were sampled at uniform random from the first 20 epochs of the chosen expert's training run. Since weights were recorded every 50 iterations, from each checkpoint a single stochastic gradient descent iteration was run with both the clean minibatch and the poisoned minibatch (as a proxy for the actual expert minibatch step) and the loss computed accordingly. Both gradient steps adhered to the training hyperparameters described above. The algorithm was run for $N = 25$ iterations through the entire dataset.

To initialize $\tilde{\ell}$, we use the original one-hot labels $y$ scaled by a temperature parameter $C$. For sufficiently large $C$, the two gradient steps in Fig. 3 will be very similar except for the changes in the poisoned examples, leading to a low initial value of $\mathcal{L}_{\text{param}}$. However if $C$ is too large, we suffer from vanishing gradients of the softmax. Therefore $C$ must be chosen to balance these two concerns.

## B.4 Compute

All of our experiments were done on a computing cluster containing NVIDIA A40 and 2080ti GPUs with tasks split roughly evenly between the two. To compute all of our numbers (averaged over 10 user models) we ended up computing approximately 3490 user models, 160 sets of labels, and 955 expert models. Averaging over GPU architecture, dataset, and model architecture, we note that each set of labels takes around 40 minutes to train. Meanwhile, each expert model takes around 10 minutes to train (fewer epochs with a more costly weight-saving procedure) and each user model takes around 40. This amounts to a total of approximately 2595 GPU-hours.

We note that the number of GPU-hours for an adversary to pull off this attack is likely significantly lower since they would need to compute as few as a single expert model (10 minutes) and a set of labels (40 minutes). This amounts to just under one GPU-hour given our setup (subject to hardware), a surprisingly low sum for an attack of this high potency.

## C   Complete experimental results

In this section, we provide expanded versions of the key tables and figures in the main text complete with standard errors as well as some additional supplementary materials. As in the main text, we compute our numbers via a three step process: (1) we start by training 5 sets of synthetic labels for each (dataset, expert model architecture, trigger) tuple, (2) we then aggregate each set of labels, and (3) we finish by training 10 user models on each interpolation of the aggregated labels and the ground truths.

For our FLIP experiments in Appendix C.1, labels are aggregated as described in Section 2 varying the number of flipped labels. Meanwhile, for our softFLIP results in Appendix C.3, we aggregate as in Section 4 by taking the average logits for each image and linearly interpolating them on parameter $\alpha$ with the ground-truth labels.

### C.1   Main results on FLIP

Fig. 2, Fig. 5a, and Table 1 showcase FLIP's performance when compared to the inner-product-based baseline and in relation to changes in dataset, model architecture, and trigger. We additionally present the raw numbers for the dot-product baseline.

| | | | 0 | 150 | 300 | 500 | 1000 | 1500 |
|---|---|---|---|---|---|---|---|---|
| C10 | r32 | s | 92.38 (0.1)/00.1 (0.0) | 92.26 (0.1)/12.4 (1.8) | 92.09 (0.1)/54.9 (2.4) | 91.73 (0.1)/87.2 (1.3) | 90.68 (0.1)/99.4 (0.2) | 89.87 (0.1)/99.8 (0.1) |
| | | t | 92.57 (0.1)/00.0 (0.0) | 92.37 (0.0)/28.4 (4.9) | 92.03 (0.0)/95.3 (1.5) | 91.59 (0.1)/99.6 (0.2) | 90.80 (0.1)/99.5 (0.3) | 89.91 (0.1)/99.9 (0.1) |
| | | p | 92.52 (0.1)/00.0 (0.0) | 92.24 (0.1)/03.3 (0.2) | 91.67 (0.0)/06.0 (0.2) | 91.24 (0.1)/10.8 (0.3) | 90.00 (0.1)/21.2 (0.3) | 88.92 (0.1)/29.9 (0.8) |
| | rl8 | s | 94.09 (0.1)/00.0 (0.0) | 94.13 (0.1)/13.1 (2.0) | 93.94 (0.1)/32.2 (2.6) | 93.55 (0.1)/49.0 (3.1) | 92.73 (0.1)/81.2 (2.7) | 92.17 (0.1)/82.4 (2.6) |
| | vgg | s | 93.00 (0.1)/00.0 (0.0) | 92.85 (0.1)/02.3 (0.2) | 92.48 (0.1)/09.2 (0.7) | 92.16 (0.1)/21.4 (0.8) | 91.11 (0.2)/48.0 (1.0) | 90.44 (0.1)/69.5 (1.6) |
| C100 | r32 | s | 78.96 (0.1)/00.2 (0.1) | 78.83 (0.1)/08.2 (0.6) | 78.69 (0.1)/24.7 (1.3) | 78.52 (0.1)/45.4 (1.9) | 77.61 (0.1)/82.2 (2.5) | 76.64 (0.1)/95.2 (0.3) |
| | rl8 | s | 82.67 (0.2)/00.1 (0.0) | 82.87 (0.1)/11.9 (0.8) | 82.48 (0.2)/29.9 (3.1) | 81.91 (0.2)/35.8 (3.1) | 81.25 (0.1)/81.9 (1.5) | 80.28 (0.3)/95.3 (0.5) |
| TI | rl8 | s | 61.61 (0.2)/00.0 (0.0) | 61.47 (0.2)/10.6 (0.9) | 61.23 (0.1)/31.6 (0.9) | 61.25 (0.2)/56.0 (1.4) | 61.45 (0.2)/51.8 (2.0) | 60.94 (0.1)/57.0 (1.5) |

Table 2: An expanded version of Table 1 in which each point is averaged over 10 user training runs and standard errors are shown in parentheses.

| baselines | 500 | 1000 | 1500 | 2500 | 5000 |
|---|---|---|---|---|---|
| inner product | 91.59 (0.1)/09.2 (0.5) | 90.70 (0.1)/17.9 (1.7) | 89.76 (0.1)/32.0 (1.6) | 87.84 (0.1)/51.3 (1.4) | 82.84 (0.1)/94.8 (0.2) |
| Random | 91.71 (0.1)/07.7 (0.8) | 90.95 (0.2)/16.2 (1.7) | 90.04 (0.1)/23.0 (1.3) | 87.64 (0.2)/42.9 (1.5) | 82.97 (0.0)/94.9 (0.2) |

Table 3: Under the scenario where $y_{\text{target}} = 9$, we show raw numbers for the inner product baseline and random selection baseline as presented in Fig. 2. The baseline was run on ResNet-32s with comparisons to the sinusoidal trigger. Each point is averaged over 10 user training runs and standard errors are shown in parentheses.

| 500 | 1000 | 2500 | 5000 | 10000 | 15000 | 20000 |
|---|---|---|---|---|---|---|
| 91.90 (0.1)/01.2 (0.1) | 91.31 (0.1)/02.7 (0.2) | 88.87 (0.1)/10.5 (0.8) | 84.80 (0.1)/30.8 (2.6) | 76.09 (0.1)/59.5 (4.5) | 66.34 (0.3)/82.8 (1.7) | 56.53 (0.3)/91.1 (1.7) |

Table 4: Under the scenario where $y_{\text{target}}$ is all but target class, we show raw numbers for the inner product baseline. The baseline was run on ResNet-32s with comparisons to the sinusoidal trigger. Each point is averaged over 10 user training runs and standard errors are shown in parentheses.

### C.2   Robustness of FLIP

We provide experimental validations of robustness of FLIP attacks.

#### C.2.1   Varying model architecture and optimizer

The attacker's strategy in our previous experiments was to train expert models to mimic exactly the user architecture and optimizer setup of the user. However, it remained unclear whether the attack would generalize if the user, for instance, opted for a smaller model than expected. As such, we looked at varying (1) model architecture and (2) optimizer between expert and user setups. For (2)

we use SGD for the expert models and Adam [46] for the user. We additionally analyze what happens when both are different.

As Table 5 indicates, the attack still performs well when information is limited. When varying optimizer, we found that CTA dropped, but, interestingly, the PTA for the ResNet-18 case was almost uniformly higher. We found a similar trend for upstream ResNet-32s and downstream ResNet-18s when varying model architecture. Surprisingly, the strongest PTA numbers for budgets higher than 150 across all experiments with a downstream ResNet-18 were achieved when the attacker used *a different expert model and optimizer*. The FLIP attack is robust to varied architecture and optimizer.

|  |  | 150 | 300 | 500 | 1000 | 1500 |
|---|---|---|---|---|---|---|
| (1) | r18 → r32 | 92.44/19.2 | 92.16/53.3 | 91.84/74.5 | 90.80/92.9 | 90.00/95.2 |
|  | r32 → r18 | 93.86/04.8 | 93.89/36.0 | 93.56/60.0 | 92.76/86.9 | 91.75/98.0 |
| (2) | r32 → r32 | 90.79/11.8 | 90.50/43.2 | 90.08/80.6 | 89.45/97.5 | 88.33/99.0 |
|  | r18 → r18 | 93.17/20.3 | 93.08/47.0 | 92.94/65.6 | 91.91/89.9 | 91.16/93.1 |
| (1 + 2) | r18 → r32 | 90.86/12.3 | 90.57/40.9 | 90.28/59.7 | 89.39/83.0 | 88.59/89.2 |
|  | r32 → r18 | 93.32/09.4 | 93.05/52.9 | 92.70/85.3 | 91.72/99.2 | 90.93/99.7 |

Table 5: FLIP performs well even when the expert and user (1) model architectures and (2) optimizers are different. Experiments are computed on CIFAR-10 using the sinusoidal trigger. Each row denotes the CTA/PTA pairs averaged over 10 experiments. The second column is structured as follows: expert → user.

Table 5 investigates whether an attacker needs to know the architecture or optimizer of the user's model. The experiments are done in the same style as Appendix C.1.

|  | 150 | 300 | 500 | 1000 | 1500 |
|---|---|---|---|---|---|
| r18 → r32 | 92.44 (0.1)/19.2 (1.3) | 92.16 (0.1)/53.3 (3.0) | 91.84 (0.0)/74.5 (2.2) | 90.80 (0.1)/92.9 (0.8) | 90.00 (0.1)/95.2 (0.6) |
| r32 → r32 | 92.26 (0.1)/12.4 (1.8) | 92.09 (0.1)/54.9 (2.4) | 91.73 (0.1)/87.2 (1.3) | 90.68 (0.1)/99.4 (0.2) | 89.87 (0.1)/99.8 (0.1) |
| r32 → r18 | 93.86 (0.1)/04.8 (0.8) | 93.89 (0.1)/36.0 (5.4) | 93.56 (0.1)/60.0 (6.6) | 92.76 (0.1)/86.9 (2.6) | 91.75 (0.1)/98.0 (0.8) |
| r18 → r18 | 94.13 (0.1)/13.1 (2.0) | 93.94 (0.1)/32.2 (2.6) | 93.55 (0.1)/49.0 (3.1) | 92.73 (0.1)/81.2 (2.7) | 92.17 (0.1)/82.4 (2.6) |
| r32 → vgg | 92.76 (0.1)/02.7 (0.1) | 92.67 (0.0)/10.0 (0.4) | 92.28 (0.1)/23.1 (1.2) | 91.41 (0.1)/47.5 (1.0) | 90.63 (0.1)/63.0 (1.5) |
| vgg → vgg | 92.85 (0.1)/02.3 (0.2) | 92.48 (0.1)/09.2 (0.7) | 92.16 (0.1)/21.4 (0.8) | 91.11 (0.2)/48.0 (1.0) | 90.44 (0.1)/69.5 (1.6) |

Table 6: An expanded version of Table 5 (1) in which each point is averaged over 10 runs and standard errors are shown in parentheses. We additionally compare the performance directly to the non-model-mixed case.

Table 6 shows more experimental results with mismatched architectures between the attacker and the user. Attacks on VGG need more corrupted examples to achieve successful backdoor attack.

|  |  | 150 | 300 | 500 | 1000 | 1500 |
|---|---|---|---|---|---|---|
| r32 | s | 90.79 (0.1)/11.8 (1.6) | 90.50 (0.1)/43.2 (3.8) | 90.08 (0.1)/80.6 (2.2) | 89.45 (0.1)/97.5 (0.3) | 88.33 (0.1)/99.0 (0.3) |
|  | t | 90.77 (0.1)/08.4 (1.3) | 90.46 (0.1)/65.0 (6.8) | 90.00 (0.1)/72.7 (5.7) | 89.07 (0.1)/98.2 (1.1) | 88.23 (0.1)/95.8 (2.5) |
|  | p | 90.60 (0.0)/03.0 (0.3) | 90.21 (0.1)/05.5 (0.3) | 89.61 (0.1)/11.1 (0.7) | 88.55 (0.1)/19.5 (0.6) | 87.39 (0.1)/31.6 (0.8) |
| r18 | s | 93.17 (0.1)/20.3 (2.2) | 93.08 (0.1)/47.0 (2.6) | 92.94 (0.0)/65.6 (1.6) | 91.91 (0.1)/89.9 (1.0) | 91.16 (0.1)/93.1 (0.7) |

Table 7: An expanded version of Table 5 (2) in which each point is averaged over 10 runs and standard errors are shown in parentheses. We additionally evaluate different choices of trigger with ResNet-32s.

|  | 150 | 300 | 500 | 1000 | 1500 |
|---|---|---|---|---|---|
| r32 → r18 | 93.32 (0.1)/09.4 (1.5) | 93.05 (0.1)/52.9 (3.0) | 92.70 (0.1)/85.3 (1.7) | 91.72 (0.1)/99.2 (0.2) | 90.93 (0.1)/99.7 (0.1) |
| r18 → r32 | 90.86 (0.1)/12.3 (1.3) | 90.57 (0.1)/40.9 (3.3) | 90.28 (0.1)/59.7 (2.6) | 89.39 (0.1)/83.0 (1.7) | 88.59 (0.1)/89.2 (0.7) |

Table 8: An expanded version of Table 5 (1+2) in which each point is averaged over 10 runs and standard errors are shown in parentheses.

### C.2.2   Number of experts $E$ and number of epochs $K$

Although for all of the above experiments we used $E = 50$ experts to create our labels, PTA is robust against using smaller number of experts. As we show in Table 9 and Fig. 6a, an attack is successful with as few as a single expert model.

| E | 150 | 300 | 500 | 1000 | 1500 |
|---|-----|-----|-----|------|------|
| 1 | 92.38 (0.1)/09.9 (0.5) | 92.05 (0.0)/45.4 (2.4) | 91.59 (0.0)/80.7 (2.7) | 90.80 (0.1)/98.0 (0.2) | 89.90 (0.1)/99.5 (0.1) |
| 5 | 92.36 (0.0)/11.9 (1.1) | 92.05 (0.1)/45.3 (3.0) | 91.42 (0.1)/86.6 (1.2) | 90.81 (0.1)/99.1 (0.2) | 89.94 (0.0)/99.8 (0.1) |
| 10 | 92.39 (0.1)/15.1 (1.6) | 92.09 (0.1)/59.7 (2.3) | 91.74 (0.0)/88.5 (1.5) | 90.91 (0.1)/99.4 (0.1) | 90.03 (0.1)/99.8 (0.1) |
| 25 | 92.33 (0.1)/10.9 (1.1) | 92.06 (0.1)/50.9 (2.2) | 91.74 (0.1)/88.9 (1.2) | 90.92 (0.1)/98.9 (0.2) | 90.03 (0.0)/99.7 (0.0) |
| 50 | 92.44 (0.0)/12.0 (1.6) | 91.93 (0.1)/54.4 (3.1) | 91.55 (0.1)/89.9 (1.1) | 90.91 (0.1)/99.6 (0.1) | 89.73 (0.1)/99.9 (0.0) |

Table 9: Understanding the effect of the number of expert models on the backdoor attack. The experiments were conducted using ResNet-32s on CIFAR-10 poisoned by the sinusoidal trigger. Standard errors in the parentheses are averaged over 10 runs.

At $K = 0$ epoch, FLIP uses an expert with random weights to find the examples to corrupt, and hence the attack is weak. In our experiments we use $K = 20$. Table 10 shows that the attack is robust in the choice of $K$ and already achieves strong performance with $K = 1$ epoch of expert training.

| K | 100 | 150 | 200 | 250 | 500 | 1000 | 1500 |
|---|-----|-----|-----|-----|-----|------|------|
| 0 | 92.38 (0.1)/00.8 (0.1) | 92.38 (0.1)/01.4 (0.2) | 92.27 (0.1)/01.9 (0.2) | 92.22 (0.1)/03.1 (0.3) | 91.79 (0.1)/05.9 (0.3) | 90.92 (0.1)/12.1 (0.9) | 89.40 (0.1)/22.4 (1.0) |
| 1 | 92.44 (0.1)/09.4 (1.1) | 92.33 (0.1)/17.0 (1.3) | 92.18 (0.1)/29.8 (2.3) | 92.15 (0.0)/43.8 (1.8) | 91.59 (0.1)/66.1 (1.7) | 90.73 (0.1)/85.0 (0.7) | 89.62 (0.1)/86.8 (1.0) |
| 5 | 92.35 (0.1)/04.6 (0.4) | 92.27 (0.0)/12.7 (1.0) | 92.22 (0.0)/22.3 (2.0) | 92.09 (0.0)/42.5 (3.2) | 91.66 (0.0)/90.3 (0.7) | 90.72 (0.1)/98.0 (0.5) | 89.80 (0.1)/99.6 (0.1) |
| 10 | 92.33 (0.0)/04.4 (0.3) | 92.38 (0.0)/09.6 (0.9) | 92.27 (0.1)/26.1 (2.2) | 92.05 (0.1)/39.4 (2.6) | 91.65 (0.0)/89.9 (1.4) | 90.67 (0.0)/99.5 (0.1) | 89.81 (0.1)/99.8 (0.1) |
| 20 | 92.54 (0.1)/05.5 (0.4) | 92.26 (0.1)/12.4 (1.8) | 92.22 (0.1)/12.1 (1.6) | 91.90 (0.1)/12.8 (1.3) | 91.73 (0.1)/87.2 (1.3) | 90.68 (0.1)/99.4 (0.2) | 89.87 (0.1)/99.8 (0.1) |
| 30 | 92.45 (0.0)/04.5 (0.5) | 92.31 (0.0)/10.9 (0.8) | 92.22 (0.1)/15.4 (0.7) | 92.25 (0.1)/34.4 (2.9) | 91.83 (0.1)/91.5 (0.8) | 90.94 (0.1)/99.3 (0.1) | 90.00 (0.0)/99.9 (0.0) |
| 40 | 92.43 (0.1)/04.4 (0.4) | 92.37 (0.1)/09.0 (0.7) | 92.26 (0.1)/19.7 (2.0) | 92.21 (0.1)/28.4 (2.1) | 91.69 (0.0)/87.0 (1.8) | 90.92 (0.0)/97.3 (0.9) | 90.01 (0.1)/99.6 (0.1) |
| 50 | 92.37 (0.1)/03.7 (0.3) | 92.41 (0.1)/08.1 (0.7) | 92.29 (0.0)/18.4 (2.2) | 92.15 (0.1)/29.1 (2.4) | 91.72 (0.1)/93.1 (1.3) | 90.87 (0.1)/99.4 (0.1) | 90.03 (0.1)/99.8 (0.1) |

Table 10: CTA/PTA pairs at different values of $K$ (the number of epochs each expert is run for) and numbers of flipped labels. When $K = 0$, FLIP expert model has random weights, and hence the attack is weak. Experiments are computed on CIFAR-10 using the sinusoidal trigger. Each point is averaged over 10 runs and standard errors are shown in parentheses.

### C.2.3   A single class source vs. multi-class source

While the majority of our paper focuses on the canonical single-source backdoor attack where $y_{\text{source}} = 9$ is, as described in Section 1.1, fixed to a single class, in this section, we consider a many-to-one attack that removes this restriction. In particular, we have that $y_{\text{source}} = \{0, ...\} \setminus \{y_{\text{target}} = 4\}$, allowing the attacker to poison images from *any class* to yield the desired prediction of $y_{\text{target}}$. We observe that this class of attack is effective as demonstrated in Table 11 and Fig. 5c.

| $y_{\text{source}}$ | 150 | 300 | 500 | 1000 | 1500 |
|---------------------|-----|-----|-----|------|------|
| all but class 4 | 92.36 (0.0)/17.2 (0.3) | 92.14 (0.1)/28.0 (1.5) | 91.67 (0.1)/58.1 (3.0) | 90.79 (0.1)/88.9 (0.9) | 90.14 (0.1)/95.6 (0.4) |
| class 9 = 'truck' | 92.26 (0.1)/12.4 (1.8) | 92.09 (0.1)/54.9 (2.4) | 91.73 (0.1)/87.2 (1.3) | 90.68 (0.1)/99.4 (0.2) | 89.87 (0.1)/99.8 (0.1) |

Table 11: FLIP is effective even when the threat model allows for (all) classes to be poisoned. Experts are trained on data where images from each class were poisoned. Experiments are computed on CIFAR-10 using the sinusoidal trigger. Each point is averaged over 10 runs and standard errors are shown in parentheses.

### C.2.4   Limited access to dataset

In this section, we consider the scenario in which the attacker is not provided the user's entire dataset (i.e., the user only distills a portion of their data or only needs chunk of their dataset labeled). As we show in Table 12, with enough label-flips, FLIP attack gracefully degrades as the knowledge of the user's training dataset decreases.

| % | 150 | 300 | 500 | 1000 | 1500 |
|---|---|---|---|---|---|
| 20 | 92.26 (0.0)/06.3 (1.1) | 92.05 (0.1)/07.2 (0.6) | 91.59 (0.1)/10.9 (0.9) | 90.69 (0.1)/15.7 (0.7) | 89.78 (0.0)/21.8 (1.2) |
| 40 | 92.31 (0.1)/10.2 (0.9) | 92.12 (0.1)/28.7 (1.6) | 91.79 (0.1)/45.2 (2.7) | 90.90 (0.1)/62.5 (1.5) | 90.08 (0.1)/74.1 (2.6) |
| 60 | 92.31 (0.0)/14.0 (0.8) | 91.99 (0.1)/45.8 (3.3) | 91.70 (0.0)/68.4 (2.7) | 90.75 (0.1)/85.5 (1.5) | 89.88 (0.1)/92.4 (0.9) |
| 80 | 92.48 (0.1)/14.0 (1.4) | 92.02 (0.1)/42.5 (3.2) | 91.70 (0.1)/80.0 (2.0) | 90.92 (0.1)/96.6 (0.4) | 89.98 (0.1)/98.5 (0.3) |

Table 12: FLIP still works with limited access to the dataset. In this setting, the attacker is provided 20%, 40%, 60%, or 80% of the user's dataset but the user's model is evaluated on the *entire dataset*. Experiments are computed on CIFAR-10 using the sinusoidal trigger. Each point is averaged over 10 runs and standard errors are shown in parentheses.

## C.3  Main results on softFLIP for knowledge distillation

We define softFLIP with a parameter $\alpha \in [0, 1]$ as, for each image, an interpolation between the soft label (i.e., a vector in the simplex over the classes) that is given by the second step of FLIP and the ground truths one-hot encoded label of that image. Clean label corresponds to $\alpha = 1$. Fig. 7 showcases softFLIP in comparison to the one-hot encoded corruption of FLIP. An expanded version with standard errors is given in Table 13. As a comparison, we show the CTA-PTA trade-off using an attack with a rounded version of softFLIP's corrupted labels such that all poisoned labels are one-hot encoded in Table 14.

| | | 0.0 | 0.2 | 0.4 | 0.6 | 0.8 | 0.9 |
|---|---|---|---|---|---|---|---|
| r32 | s | 90.04 (0.1)/100. (0.0) | 90.08 (0.0)/100. (0.0) | 90.11 (0.1)/100. (0.0) | 90.45 (0.1)/99.9 (0.0) | 91.02 (0.0)/99.0 (0.1) | 91.95 (0.0)/25.3 (3.0) |
| | t | 88.05 (0.1)/100. (0.0) | 88.43 (0.1)/100. (0.0) | 88.44 (0.1)/100. (0.0) | 88.99 (0.1)/100. (0.0) | 89.86 (0.1)/100. (0.0) | 90.85 (0.0)/100. (0.0) |
| | p | 88.02 (0.1)/44.5 (0.4) | 88.26 (0.1)/41.9 (0.4) | 88.62 (0.1)/38.8 (0.5) | 89.10 (0.1)/31.1 (0.4) | 91.64 (0.1)/05.1 (0.3) | 92.04 (0.1)/00.1 (0.0) |
| r18 | s | 92.97 (0.1)/98.7 (0.3) | 92.92 (0.1)/97.3 (1.3) | 93.13 (0.1)/96.0 (2.1) | 93.25 (0.1)/95.6 (0.9) | 93.67 (0.1)/86.1 (1.4) | 93.91 (0.1)/33.6 (4.9) |

Table 13: CTA-PTA trade-off for softFLIP with varying $\alpha \in \{0.0, 0.2, 0.4, 0.6, 0.8, 0.9\}$, varying architecture (ResNet-32 and ResNet-18), and varying trigger patterns (sinusoidal, Turner, pixel). Each point is averaged over 10 runs and standard errors are shown in parentheses.

| | | 0.0 | 0.2 | 0.4 | 0.6 | 0.8 | 0.9 |
|---|---|---|---|---|---|---|---|
| r32 | s | 89.82 (0.1)/100. (0.0) | 89.78 (0.1)/99.9 (0.0) | 90.00 (0.1)/99.9 (0.0) | 90.25 (0.1)/99.9 (0.0) | 91.08 (0.1)/99.0 (0.2) | 92.21 (0.1)/29.2 (2.9) |
| | t | 87.30 (0.1)/99.6 (0.4) | 87.47 (0.1)/99.8 (0.1) | 87.83 (0.1)/100. (0.0) | 88.52 (0.1)/100. (0.0) | 89.76 (0.1)/99.3 (0.4) | 91.27 (0.1)/100. (0.0) |
| | p | 87.88 (0.1)/42.1 (0.6) | 88.09 (0.1)/39.7 (0.3) | 88.45 (0.1)/36.5 (0.8) | 89.22 (0.1)/29.9 (0.4) | 91.53 (0.1)/07.9 (0.3) | 92.53 (0.0)/00.1 (0.0) |
| r18 | s | 92.59 (0.1)/95.2 (1.1) | 92.82 (0.1)/92.6 (2.1) | 92.87 (0.1)/89.7 (1.5) | 93.25 (0.0)/90.0 (2.1) | 93.56 (0.1)/57.3 (2.5) | 94.12 (0.1)/03.9 (0.4) |

Table 14: A rounded version of Table 13, the results are averaged over 10 runs, and the standard errors are shown in parentheses.

# D  Additional experiments

## D.1  Sparse regression approach with $\ell_1$ regularization

FLIP performs an approximate subset selection (for the subset to be label-corrupted examples) by solving a real-valued optimization and selecting those with highest scores. In principle, one could instead search for sparse deviation from the true labels using $\ell_1$ regularization, as in sparse regression. To that end we experimented with adding the following regularization term to $\mathcal{L}_{\mathrm{param}}$ in Section 2:

$$\frac{\lambda}{|\tilde{B}_k^{(j)}|} \sum_{x \in \tilde{B}_k^{(j)}} \left\| y_x - \mathrm{softmax}(\hat{\ell}_x) \right\|_1 , \tag{4}$$

where $y_x$ refers to the one-hot ground-truth labels for image $x$. Table 15 shows that this approach has little-to-no improvement over the standard FLIP (which corresponds to $\lambda = 0$).

## D.2  FLIP against defenses

We test the performance of FLIP when state-of-the-art backdoor defenses are applied. We evaluate FLIP on CIFAR-10 with all three trigger types on three popular defenses: kmeans [15], PCA [89],

| $\lambda$ | 150 | 300 | 500 | 1000 | 1500 |
|---|---|---|---|---|---|
| 0. | 92.27 (0.1)/16.2 (1.2) | 92.05 (0.0)/59.9 (3.0) | 91.48 (0.1)/90.6 (1.5) | 90.75 (0.1)/99.5 (0.1) | 89.89 (0.0)/99.9 (0.0) |
| 0.5 | 92.31 (0.1)/15.4 (1.6) | 91.96 (0.0)/49.4 (2.2) | 91.64 (0.1)/90.1 (1.5) | 90.62 (0.1)/99.6 (0.1) | 89.77 (0.0)/99.8 (0.0) |
| 1. | 92.29 (0.1)/11.5 (0.8) | 91.96 (0.0)/44.3 (2.7) | 91.72 (0.1)/90.8 (0.9) | 90.70 (0.1)/98.7 (0.1) | 89.87 (0.1)/99.8 (0.0) |
| 2. | 92.35 (0.1)/13.6 (1.6) | 92.02 (0.0)/47.9 (2.3) | 91.68 (0.0)/94.1 (1.0) | 90.76 (0.1)/99.2 (0.1) | 89.83 (0.1)/99.7 (0.1) |
| 3. | 92.19 (0.1)/09.0 (0.6) | 91.85 (0.1)/51.1 (1.8) | 91.46 (0.1)/88.0 (2.0) | 90.70 (0.1)/99.1 (0.1) | 89.75 (0.1)/99.6 (0.1) |
| 4. | 92.32 (0.1)/13.1 (1.4) | 92.01 (0.0)/57.4 (2.2) | 91.54 (0.1)/89.5 (0.8) | 90.57 (0.0)/99.3 (0.1) | 89.75 (0.1)/99.6 (0.1) |
| 5. | 92.24 (0.1)/15.0 (1.6) | 92.02 (0.0)/53.3 (2.8) | 91.57 (0.1)/86.1 (1.6) | 90.45 (0.1)/99.3 (0.1) | 89.53 (0.0)/99.7 (0.0) |

Table 15: $\ell_1$-regularization (Eq. (4)) on FLIP does not improve performance. We note that when $\lambda = 0$, FLIP is as introduced previously. Experiments are computed on CIFAR-10 using the sinusoidal trigger. Each point is averaged over 10 runs and standard errors are shown in parentheses.

and SPECTRE [37]. We find that SPECTRE is quite effective in mitigating FLIP, whereas the other two defenses fail on the periodic and Turner triggers. This is consistent with previously reported results showing that SPECTRE is a stronger defense. We emphasize that even strictly stronger attacks that are allowed to corrupt the images fail against SPECTRE. In any case, we hope that our results will encourage practitioners to adopt strong security measures such as SPECTRE in practice, even under the crowd-sourcing and distillation settings with clean images. In our eyes, finding strong backdoor attacks that can bypass SPECTRE is a rewarding future research direction.

| | | 150 | 300 | 500 | 1000 | 1500 |
|---|---|---|---|---|---|---|
| $s$ | kmeans | 92.28 (0.0)/10.8 (1.3) | 92.15 (0.0)/55.6 (3.4) | 91.68 (0.1)/84.8 (1.2) | 90.78 (0.0)/96.3 (0.7) | 90.42 (0.1)/86.3 (8.4) |
| | PCA | 92.34 (0.1)/11.7 (1.2) | 91.95 (0.1)/58.8 (3.6) | 91.54 (0.1)/85.3 (1.8) | 90.84 (0.1)/98.4 (0.3) | 90.40 (0.1)/79.4 (7.4) |
| | SPECTRE | 92.50 (0.1)/00.2 (0.0) | 92.55 (0.0)/00.2 (0.0) | 92.43 (0.1)/00.2 (0.0) | 92.06 (0.1)/01.3 (0.1) | 91.47 (0.1)/01.7 (0.3) |
| $p$ | kmeans | 92.13 (0.1)/02.7 (0.2) | 91.82 (0.1)/04.9 (0.3) | 91.36 (0.1)/08.3 (0.4) | 92.37 (0.2)/01.5 (1.4) | 88.60 (0.1)/30.4 (0.4) |
| | PCA | 92.14 (0.1)/02.7 (0.2) | 91.83 (0.0)/05.2 (0.1) | 91.73 (0.2)/05.9 (1.3) | 92.26 (0.1)/02.1 (0.9) | 92.09 (0.2)/02.5 (1.6) |
| | SPECTRE | 92.57 (0.1)/00.0 (0.0) | 92.42 (0.1)/00.1 (0.0) | 92.54 (0.0)/00.0 (0.0) | 92.34 (0.1)/00.1 (0.0) | 92.27 (0.0)/00.1 (0.0) |
| $t$ | kmeans | 92.32 (0.1)/21.2 (4.4) | 92.06 (0.1)/86.5 (7.0) | 91.70 (0.1)/95.9 (1.8) | 90.75 (0.1)/96.0 (2.2) | 90.01 (0.1)/96.4 (2.7) |
| | PCA | 92.25 (0.0)/36.8 (6.7) | 91.97 (0.1)/95.2 (1.7) | 91.63 (0.1)/96.8 (1.8) | 90.79 (0.1)/99.6 (0.1) | 89.95 (0.1)/98.3 (0.5) |
| | SPECTRE | 92.42 (0.1)/00.1 (0.0) | 92.36 (0.1)/00.1 (0.0) | 92.17 (0.0)/00.3 (0.1) | 91.45 (0.0)/00.7 (0.4) | 90.70 (0.1)/03.4 (2.6) |

Table 16: SPECTRE [37] is mitigates our attack when armed with each of our triggers. Experiments are computed on CIFAR-10. Each point is averaged over 10 runs and standard errors are shown in parentheses.

### D.3 Fine-tuning large pretrained ViTs

In this section we show that FLIP is robust under the fine-tuning scenario. In particular, both ResNet- and VGG-trained labels successfully backdoor Vision Transformers (ViTs) [25] pre-trained on ImageNet1K where all layers but the weights of classification heads are frozen. When an expert is trained on ResNet-32 and the user fine-tines a model on the FLIPed data starting from a pretrained ViT, the attack remains quite strong despite the vast difference in the architecture and the initialization of the models. The same holds for an expert using VGG-19.

| | 150 | 300 | 500 | 1000 | 1500 |
|---|---|---|---|---|---|
| r32 $\rightarrow$ vit (FT) | 95.42 (0.0)/01.6 (0.1) | 95.29 (0.0)/06.4 (0.1) | 95.06 (0.0)/14.5 (0.2) | 94.67 (0.0)/31.1 (0.3) | 94.27 (0.1)/40.2 (0.3) |
| vgg $\rightarrow$ vit (FT) | 95.40 (0.0)/01.4 (0.1) | 95.31 (0.0)/06.9 (0.1) | 95.07 (0.0)/16.7 (0.1) | 94.64 (0.1)/30.2 (0.1) | 93.98 (0.0)/30.7 (0.2) |

Table 17: ResNet and VGG-19 learned corrupted labels successfully poison ImageNet-pretrained ViT models in the fine-tuning scenario.

