# OpenReview forum: "Label Poisoning is All You Need"
_NeurIPS.cc/2023/Conference — NeurIPS 2023 poster_

### Official Review · Reviewer_ewXv · 2023-07-03

**Soundness:** 3 good
**Presentation:** 3 good
**Contribution:** 3 good
**Rating:** 5
**Confidence:** 4

**Summary:**

This work introduces a novel backdoor attack, (soft)FLIP, which only requires modifying the labels of the training samples and supports arbitrary choice of triggers. (soft)FLIP is based on a trajectory-matching algorithm, which optimizes the poison labels by simulating how normal backdoor poison samples affect the model parameters during training. The experimental results validate the high effectiveness of (soft)FLIP on CIFAR10 and CIFAR100, across two ResNet architectures, three different types of triggers, and different numbers of poison samples.

**Strengths:**

1. The idea of backdoor poisoning only modifying the label is rather novel and interesting. Meanwhile, the reported attack results look quite promising.
2. The motivation of label-only poisoning backdoor attack is well supported by two real-world scenarios (crowd-sourcing image annotation & knowledge distillation). Overall, the paper is well structured.
3. The time consumption to conduct FLIP could go less than 1 GPU hour, making the proposed attack more practical.

**Weaknesses:**

1. My major concern is about the limitation of the model architectures you investigated. Through both the main paper and the appendix, you only investigate ResNet architectures (r18 & r32), which only cover a small spectrum of CV deep models. The lack of model architecture diversity subsequently leads to readers' doubt --- whether your attack could generalize well in the real world, on more recent architectures which are not in the ResNet family. I think for the paper's acceptance, it's necessary to consider other types of model architectures (e.g. VGG, Vision Transformer).
2. Another concern I have is the lack of studies from the defender's perspective. On the one hand, there are many existing advanced poison-cleanser defenses (e.g. [1] and [2]). I suggest including a brief study about whether your poisoning attack is resistant to these defenses in your paper's main body. It would be even better if you provide some insights into potential defenses that may effectively resist FLIP.
3. The logic in paragraph 2 (Line 30-36) may need refining (e.g. "However" in Line 32 is inappropriate).
4. Line 100-101 the claim "the corrupted labels chosen by FLIP do not degrade the clean accuracy of the model" should be diminished, since FLIP still leads to some CTA drop.
5. The caption for Figure 5(a) is wrong, please correct it.
6. May need to add identifiers (e.g. number of experts & number of poisoning samples) in Table 4.

I will update my score once the above concerns are addressed.

[1] Hayase, Jonathan, Weihao Kong, Raghav Somani, and Sewoong Oh. "Spectre: Defending against backdoor attacks using robust statistics." In *International Conference on Machine Learning*, pp. 4129-4139. PMLR, 2021.

[2] Tang, Di, XiaoFeng Wang, Haixu Tang, and Kehuan Zhang. "Demon in the variant: Statistical analysis of {DNNs} for robust backdoor contamination detection." In *30th USENIX Security Symposium (USENIX Security 21)*, pp. 1541-1558. 2021.

**Questions:**

1. Your attack setting requires selecting both a (coarse) source class and a target class. What about all-to-one attack (every class is the source class, which is more common and more practical)?
2. In Line 195-198, what if $C$ is small? Could you explain more about the selection and meaning of $C$?
3. Could you provide results using all three trigger styles, for CIFAR-10 (r18) and CIFAR-100 (r18 & r32) settings?
4. There are some recent backdoor poisoning attacks (e.g. [1] and [2]) that also emphasize the manipulation of poison labeling. What's the relationship between your work and them? It would be great to include a short discussion in Related Work.

[1] Tang, Di, XiaoFeng Wang, Haixu Tang, and Kehuan Zhang. "Demon in the variant: Statistical analysis of {DNNs} for robust backdoor contamination detection." In *30th USENIX Security Symposium (USENIX Security 21)*, pp. 1541-1558. 2021.

[2] Qi, Xiangyu, Tinghao Xie, Yiming Li, Saeed Mahloujifar, and Prateek Mittal. "Revisiting the assumption of latent separability for backdoor defenses." In *The eleventh international conference on learning representations*. 2022.

**Limitations:**

Limitations and potential defenses against the proposed attack are not explicitly discussed.

---

> ### Author Rebuttal · Authors · 2023-08-09
>
> We wish to thank the reviewer and address their comments point-by-point below:
> ### Weaknesses:
> 1. We refer the reader to the **Experiments on Larger Models + Transformers** section of the general rebuttal above for details on our experiments against the VGG and Vision Transformer architectures.
> 2. We refer to the **Defense** section of the general rebuttal above for details on our experiments against backdoor attacks in the literature. Of note, we recommend spectral defense methods like SPECTRE as a way to mitigate our attack.
> 3. In response to feedback, we have rewritten many of the sections in the paper and are reworking some of the ambiguity in the writing. As such we have tailored that paragraph to underscore our core contribution: the discovery of a new attack vector we believe practitioners should be aware of.
> 4. We have moderated the claim to: “minimally degrade the clean accuracy of the model”
> 5. This has been corrected to “Triggers”
> 6. In response to the suggestion, we plan to add identifiers to all of our tables.
> ### Questions:
> 1. We evaluated a version of FLIP with *no source class* on CIFAR-10 with the sinusoidal trigger and found that our method was still highly successful as detailed in the below table.
> $150$|$300$|$500$|$1000$|$1500$
> -|-|-|-|-
> $92.36/17.2$|$92.14/28.0$|$91.67/58.1$|$90.79/88.9$|$90.14/95.6$
> 2. $C$ is a temperature hyperparameter for our targets’ softmax initializations. The goal of $C$ is to distribute the labels’ gradient mass during the early stages of FLIP’s evaluation. A $C$ too high “vanishes gradients” by sending the logits of non-ground-truth classes too close to $0$. A $C$ low can lead to high volatility in the early stages of evaluation. While results don’t differ greatly with reasonable choices of $C$, we found that a well-selected value speeds up evaluation. As with most of our hyperparmeters, our selection of $C$ was done in a typical grid-search fashion.
> 3. We are working on expanding Table 1 as suggested. The final results will be added to the paper promptly.
> 4. While it is true that both papers propose nonstandard labeling of the poison data, both attacks still rely on image poisoning to succeed. The manipulation of labels is designed to supplement the traditional backdoor attack instead of replacing it. We believe this is an interesting line of work which is distinct from and complementary to the results of our paper.

---

> > ### Comment · Reviewer_ewXv · 2023-08-15
> > **Thanks for your efforts in the rebuttal**
> >
> > I appreciate the efforts the authors made during the rebuttal period, which has addressed most of my concerns. I will increase my rating to 5 for now. It would be great if in the following days, you could also 1) show some other possible experiment settings (e.g. vit -> vit, vgg -> vit) in "Experiments on Larger Models + Transformers." (even if they fail); 2) provide results regarding my 3rd concern in "Questions" about all three triggers.

---

> > > ### Author Response · Authors · 2023-08-16
> > > **Thank you.**
> > >
> > > We really appreciate the time you took in responding to our rebuttal. We are working hard to get (some of) those experiments done before Aug 21st. We will keep you posted on the results.

---

> > > > ### Author Response · Authors · 2023-08-21
> > > > **additional experiments.**
> > > >
> > > > Here are some experimental results we could get in the given time.
> > > >
> > > > This is VGG->ViT
> > > > $150$|$300$|$500$|$1000$|$1500$
> > > > -|-|-|-|-
> > > > $95.43/01.4$|$95.31/06.9$|$95.07/16.8$|$94.59/30.1$|$94.01/30.5$
> > > >
> > > > For Table 1, here is a result with Pixel trigger for CIFAR-10 and r18
> > > > ||$150$|$300$|$500$|$1000$|$1500$
> > > > -|-|-|-|-|-
> > > > $p$|$93.53/01.1$|$93.20/03.6$|$92.87/06.5$|$91.92/12.9$|$90.80/23.4$
> > > >
> > > > We are working on getting more results.

---

### Official Review · Reviewer_uwV8 · 2023-07-05

**Soundness:** 3 good
**Presentation:** 2 fair
**Contribution:** 3 good
**Rating:** 5
**Confidence:** 3

**Summary:**

The paper studies backdoor attacks, in which a subset of training examples are poisoned with the goal of flipping predicted labels at test time with minimal modifications to the test instances (e.g., by adding a small logo or invisible noise).

Previous backdoor attacks either inject malicious data or modify the instances, usually by adding a pattern that correlates with the preferred output label.

This work shows that by *only changing the labels* of a small subset of the training examples one can still launch a successful backdoor attack.

At a technical level, the attack first performs "traditional" attacks to train multiple poisoned (backdoored) models while storing certain information about such attacks regarding their trajectory of the poisoned models. Then such trajectories are used to find out a minimal set of labels whose flipping can have a close effect (to the original attacks).

Concretely, by only changing 1000 example labels in CIFAR-10, the paper can achieve accuracy under attack (i.e., successfully flipping the label to the desired one) of ~1 while the actual accuracy without attack remains ~0.9

The paper compares their (in my opinion impressive) attack with a natural (but rather weak) attack that is based on inner product, in which the labels are flipped if the trigger patter is "weakly" present in an input instance (and that is estimated using inner product between the instance and the pattern). The comparison shows that the proposed attack is significantly more effective.

**Strengths:**

The main question of the paper is a natural question, and the answer is rather surprising. I personally would not think that label change is so powerful to launch a backdoor attack.

The attack's idea is smart and could find further applications in adversarial learning by reducing attack power to label-only setting.

**Weaknesses:**

The main weakness of the paper is its poor presentation, in which many sentences and phrases are unclear for readers. Here are some examples.

* calling a dataset with flipped labels "clean data" (in line 105) is a misnomer. you can call them "clean instances" but the data is poisoned, as labels are also part of the data.

* I did not understand the exact definition of the attack in the knowledge distillation setting. Please give a formal mathematical definition.

There are many unclear sentences and phrases such as:

"we convert our continuous logits to a 170 discrete set of label flips."
"a set of training trajectories of backdoored ‘expert models"
"Intuitively, each gradient step differs only by the poisoned images"

Letter T is overloaded: it is used both for the trigger and the length of the saved trajectories.

What does "intensity and stealth" of the attack mean, formally?

What is "interpolation percentage"?






**Questions:**

Refer to the weakness comments and explain the phrases and formally define the attack model for the knowledge distillation setting. Also, what is the (time) complexity of your attack. Can you report a few numbers?

**Limitations:**

The computational complexity of the attack is not clear to me. it seems like your attack is much heavier than the "traditional" ones. but since this is an attack (not a defense) the complexity is still more OK, as there is no real symmetry between the two.

---

> ### Author Rebuttal · Authors · 2023-08-09
>
> We wish to thank the reviewer and address their comments point-by-point below:
> ### Weaknesses:
> 1. *(Poor Presentation.)* Based on the constructive feedback from our reviewers, we are revising the writing significantly.
> 2. *(Clean Instances.)* We will clarify this in the revisions.
> 3. *(Knowledge Distillation Setting.)* We will clarify our threat model for the knowledge distillation setting as follows in the revision.
>
>    *In our distillation setting, the attacker provides a teacher model to the student, which is used to (soft) label the student’s distillation set of clean images. To produce these labels, we assume the attacker has read-access to the distillation set and can construct the teacher to produce arbitrary outputs on it. The goal of the adversary is to create a backdoor in the student model after it is trained on the adversarially labeled distillation set.*
>
>    In practice, the adversary can first compute the soft labels that produce a backdoor and then overfit a teacher model to these labels such that, when queried, the computed labels are revealed.
>
>    Explicitly, let $s(\cdot; W_s)$ be the user / student model, $t(\cdot; W_t)$ be the potentially-adversarial teacher model and $X$ be the user's clean distillation images. As in the standard distillation setup, for standard distillation loss $\mathcal{L}_{\mathrm{distill}}$, the user seeks to minimize:
>
>    $$\hat{W_s} = \min_{W_s} \mathcal{L}_{\mathrm{distill}}\big(s( X ; W_s), t(X ; \hat{W}_t)\big)$$
>
>    Now, an adversarial attacker's goal is to embed a backdoor into the model to satisfy the constraints in Equation (1) and Section 1.1. However, unlike most backdoor scenarios where the adversary maintains control over the images $X$, in our setting, the attacker *can only* modify the model $t$ to select an adversarial $t(\cdot; W_t)$.
> 4. *(Unclear Sentences and Phrases.)*
>    1. We clarified what is meant by conversion in the revised text (i.e., the selection process detailed in Section 3.3).
>    2. We clarified the phrase to: “the first step of our attack is to record a set of training trajectories (i.e., the checkpoints of *expert models* trained on data corrupted as per a traditional backdoor attack with trigger $T(\cdot)$ and target $y_{\mathrm{target}}$ of interest).
>    3. We clarified the intuition that there are analagous batches of expert training data and user training data.
> 4. *(Letter T.)* The length of trajectory will be changed to $K$.
> 5. *(Intensity and Stealth.)* The intensity of our attack refers to its Poison Test Accuracy (PTA) while, the stealth refers to how hard the attack is to detect via the Clean Test Accuracy (CTA). We will revise the paper to use PTA and CTA consistently throughout.
> 6. *(Interpolation Percentage.)* The interpolation percentage $\alpha$ adjusts the PTA / CTA tradeoff by linearly interpolating between the logit-labels generated by FLIP ($\alpha = 0.0$) and the one-hot versions of the ground-truth labels of the dataset ($\alpha = 1.0$). Explicitly, as $\alpha \rightarrow 1$ the CTA increases and the PTA decreases. This definition has been clarified.
>
> ### Questions
> 1. *(Time complexity.)* While most of our experiments are run with 50 experts evaluated for 20 epochs each, as we show in Table 5 and the table below, our attack is effective with far fewer experts and epochs $K$, respectively. As such, on average, training a single expert for 20 epochs took less than 10 minutes on A40 and 2080ti GPUs. This process can obviously be parallelized for more experts. In addition, our algorithm, was run for 25 iterations which took nearly 25 minutes on the same setup. Together the just-over-half a GPU hour amounts to around a single model training run.
>
>    In terms of complexity, the label optimization has similar computational cost to training an expert, so we expect it to scale similarly to model training as the dataset and model size increases.
>
>    For more information, please refer Section A.4 of the Supplementary Material.

---

> > ### Comment · Reviewer_uwV8 · 2023-08-14
> > **Thanks for the clarifications**
> >
> > I will follow up if I have more questions. For now I do not have more.

---

> > > ### Author Response · Authors · 2023-08-16
> > > **Thank you.**
> > >
> > > Thank you for responding to our rebuttal.

---

### Official Review · Reviewer_TrAf · 2023-07-06

**Soundness:** 2 fair
**Presentation:** 3 good
**Contribution:** 2 fair
**Rating:** 4
**Confidence:** 4

**Summary:**

This paper studies an interesting problem whether backdoor attacks are effective if only manipulating the labels. They propose a method based on searching labels that can match parameters with pre-trained conventional backdoored models. An effective algorithm is shown and experiments are conducted to validate the effectiveness.

**Strengths:**

The problem itself is important and the motivation is proper. The logic of the whole paper is clear and easy to follow. The algorithm is clearly illustrated, and experiments are conducted to show the effectiveness of the proposed method.


**Weaknesses:**

1. Overall, this paper seems to be written in a rush as many details are missing.
2. Scenario two needs more demonstration. It is not clear which model(teacher or student) authors would like to poison.
3. It is not sure whether this work is the first study of label-poison backdoor, as *Clean-image Backdoor: Attacking Multi-label Models with Poisoned Labels Only* already studied a similar problem. A comparison is needed in related work, and it should be considered as a baseline.
4. Proposed method is not a novel one, because in the paper 'Stronger data poisoning attacks break data sanitization defenses', they proposed to search for poisoning samples based on a decoy model. From my perspective, the method proposed in this paper is a direct application of that method to the backdoor attack.
5. The guarantee or intuition behind the proposed matching procedure(iterations from line 212) is not clear. It is not straightforward to understand why updating poisoned labels at each checkpoint will match the parameters between expert models and label-poisoned models, because the re-training process can lead to totally different local minima.
6. Please give more details about the dot-product baseline. If it is proposed in previous works, please provide references. If it is defined in this work, show more details especially the difference between it and the proposed method.


**Questions:**

1. I am confused about the attackers' ability in this work. Are you assuming that the clean training set is also accessible to attackers? Otherwise one can not train expert models using conventional backdoor methods. Since you need to train expert models, do you assume that the images can be manipulated? If so, is this assumption too strong in practice and does it contradict the motivation?
2. Need more details about training expert models. Are you training multiple expert models using the same architecture, trigger, source class and target class? If so, what is the difference between these expert models? Besides, expert models are only trained for 20 epochs, why not train for more epochs?
3. What is the intuition behind the loss function in Eq.(2)? Give more details about why it is designed in that form.
4. Why lack the experiments facing backdoor defenses?



**Limitations:**

Mentioned in weakness and questions.

---

> ### Author Rebuttal · Authors · 2023-08-09
>
> We thank the reviewer and address their comments point-by-point below:
> ### Weaknesses:
> 1. We are adding several missing details to the revision as per reviewers’ feedback. In conjunction with other edits, we believe this has greatly improved the quality of the presentation (and experimental results) of the paper.
> 2. We will clarify our threat model for the second scenario as follows in the revision.
>
>    *In our distillation setting, the attacker provides a teacher model to the student, which is used to (soft) label the student’s distillation set of clean images. To produce these labels, we assume the attacker has read-access to the distillation set and can construct the teacher to produce arbitrary outputs on it. The goal of the adversary is to create a backdoor in the student model after it is trained on the adversarially labeled distillation set.*
> 3. While we agree with the reviewer that a label-space attack was introduced recently in [14], the setting where the attack applies is completely different, as are the resulting attack designs and priorities. In particular, [14] uses an algorithmically selected trigger combined with a straightforward corruption of the labels, while we assume the trigger is pre-specified and optimize the labels to achieve the attack.
>
>    Additionally, in [14] the trigger must be the presence of a combination of labels which appear in the data with frequency matching the desired poisoning rate, which is itself selected to balance CTA and PTA. In contrast, our attack will (in principle) work for arbitrary triggers and we demonstrate successful attacks with several triggers which are not present in any of the images in the dataset.
>
>    Finally, [14] works only for multi-label tasks and cannot be applied directly to our single label setting.
> 4. The use of a “decoy” model for efficiently solving bilevel optimization is not new and has been done in [34,11,22,49,31]. Our method for solving bilevel optimization is not novel; we tried several approaches and chose the one that worked best: trajectory matching. In particular, the method proposed in [34] is tailored for binary classification (Algorithm 4 and optimization problem (14)) and cannot be directly applied to our setting with multiple classes.
> 5. Informally, the goal of the trajectory matching is to minimize the drift away from the expert trajectory when training with only poisoned labels. Averaged over the full trajectory, this amounts to minimizing the distance between a model trained purely with poison labels and the final expert checkpoint under the assumption that they share the same randomness (initialization, batch order). While it is true that re-training with different randomness will lead to a completely different local minima, it is rare in machine learning to see changes in randomness have a large effect on the functional accuracy (CTA, PTA) of the final model. Also, to reduce overfitting to a particular random seed, we can average over multiple expert trajectories.
> 6. The dot-product baseline is our own and corresponds to flipping the $k$ images with the largest dot product with the trigger. Figure 2 shows that dot-product baseline requires an order of magnitude more poisoned examples than FLIP to successfully backdoor the trained model. Such a massive poison injection results in a rapid drop in CTA, causing an unfavorable CTA-PTA tradeoff curve.
>
>    We remark that while the dot-product computes similarity to the trigger in image space, FLIP considers a much deeper notion of similarity in the parameter space of a trained model by targets images that induce similar gradients to the trigger.
> ### Questions:
> 1. While most of our experiments assume that the attacker has access to the full training set, in the table below we show that this assumption can be relaxed (i.e., access to only a subset of the training set).
>
>    Under the two scenarios and threat model we consider, the attacker can only corrupt the labels and cannot alter any images. As long as the student training adheres to this constraint, we believe the attacker’s power is limited and does not contradict our motivation. In practice, we consider crowdsourcing and knowledge distillation, where it is natural to assume that the attacker can alter the data locally as long as only the labels are corrupted when the student model is trained.
>
>    ||$150$|$300$|$500$|$1000$|$1500$
>    -|-|-|-|-|-
>    $20$|$92.26/06.3$|$92.05/07.2$|$91.59/10.9$|$90.69/15.7$|$89.78/21.8$
>    $40$|$92.31/10.2$|$92.12/28.7$|$91.79/45.2$|$90.90/62.5$|$90.08/74.1$
>    $60$|$92.31/14.0$|$91.99/45.8$|$91.70/68.4$|$90.75/85.5$|$89.88/92.4$
>    $80$|$92.48/14.0$|$92.02/42.5$|$91.70/80.0$|$90.92/96.6$|$89.98/98.5$
> 2.  For most of our experiments we trained 50 (studied in Table 5)  expert models with identical architecture, trigger, source class, and target class. However, to promote generalization across training trajectories the random initialization was varied for each of the experts. As shown in **Experiments on $K$** in the main rebuttal, we found that small values of $K$ (i.e., the number of epochs the experts are trained for) work well since checkpoints later on in training drift away from the student model training trajectory. For more information on the experts, please refer to supplementary material Section A.
> 3. Informally, (2) seeks to optimize the labels of the user’s training dataset such that the distance between the parameters of a model trained on the user’s set and the parameters of an expert model trained on a traditionally-backdoored dataset is minimized. This goal is clearly reflected in the numerator of (2). The denominator of (2) normalizes this quantity across the entire expert trajectory. If successful, the student model trained with label-only corruption would inherit the backdoor of the expert model.
> 4. We refer to the **Defense** section of the general rebuttal above for details on our experiments against backdoor attacks in the literature.

---

> > ### Comment · Reviewer_TrAf · 2023-08-15
> >
> > Thanks for the clarification. The response address some of my concerns. Although I feel the studied problem is interesting, I still hesitate the fundamental / practical importance of the problem, beyond existing / original backdoor attacks. I would raise my score to 4 now and see the discussions of other reviewers.

---

> > > ### Author Response · Authors · 2023-08-16
> > > **Thank you.**
> > >
> > > We thank the reviewer for reading our rebuttal and engaging in the conversation with us. The major "surprise" in our work is that corrupting only the label can be successful in creating backdoors for pre-defined triggers. We believe this is quite practical in the two scenarios we considered: crowdsourced annotations and knowledge distillation, which is not covered by standard backdoor attacks.
> > >
> > > We are happy to provide any further information if there are specific remaining concerns.

---

### Official Review · Reviewer_QpmH · 2023-07-21

**Soundness:** 3 good
**Presentation:** 3 good
**Contribution:** 4 excellent
**Rating:** 4
**Confidence:** 5

**Summary:**

The paper first proposes a method that a backdoor attack can be done with only label poisoning. The authors introduce a new algorithm called FLIP that corrupts only the labels in the training set to create a backdoor attack. The first step of FLIP is to collect a set of training trajectories of backdoored ‘expert models’. Then, a set of real-valued logits are produced that induce similar parameters to the poisoned data when combined with clean images. Moreover, parameter-matching loss are proposed to update the logits. Finally, label flips are implanted by logits. What’s more, SoftFLIP is applied in the setting of knowledge distillation. The method is evaluated on extensive experiments and shows its effectiveness.

**Strengths:**

The article reveals the threats of only label poisoning which is rarely considered before. The method proposed seems correct and the experiments are sufficient. The experiments’ settings are rational and the article is well organized.

**Weaknesses:**

There are no any experiments on against backdoor defense. It is also an important aspect of evaluating the backdoor attack. The backbones are limitated to the ResNet. How about the results on transformers which are more popular backbones in knowledge distillation?

**Questions:**

In Table 1, r18 s has a poor condition on PTA, it seems strange. Do your method have the robustness against fine-tuning?

**Limitations:**

No any backdoor defenses are tested in the article.

---

> ### Author Rebuttal · Authors · 2023-08-09
>
> We wish to thank the reviewer and address their comments point-by-point below:
> ### Weaknesses:
> 1. We refer to the **Defense** section of the general rebuttal above for details on our experiments against backdoor attacks in the literature.
> 2. We refer to the **Experiments on Larger Models + Transformers** section of the general rebuttal above for details on our experiments against the VGG and Vision Transformer architectures.
> ### Questions:
> 1. To lower the statistical noise of the experiments presented in our submission we reran each experiment 7 *additional* times so that each figure is averaged over 10 experiments. In this process the non-monotonicity of ResNet-18s on CIFAR-10 (with respect to the number of label flips) was corrected. These higher confidence results can be found in Table 7 of the supplementary material along with similar updates for each experiment in the paper.
> ||$150$|$300$|$500$|$1000$|$1500$
> -|-|-|-|-|-
> CIFAR-10|$94.13/13.1$|$93.94/32.2$|$93.55/49.0$|$92.73/81.2$|$92.17/82.4$
> CIFAR-100|$82.87/11.9$|$82.48/29.9$|$81.91/35.8$|$81.25/81.9$|$80.28/95.3$
> 2. We refer to the Vision Transformer experiment in the **Experiments on Larger Models + Transformers** section of the general rebuttal above. In particular, we found that when the last few layers of a transformer pretrained on ImageNet are finetuned on our corrupted labels, the attack succeeds. This is true even when the labels are generated with experts of a different architecture.
> ### Limitations:
> 1. We refer to the **Defense** section of the general rebuttal above.

---

### Official Review · Reviewer_tDVF · 2023-07-25

**Soundness:** 2 fair
**Presentation:** 3 good
**Contribution:** 2 fair
**Rating:** 5
**Confidence:** 4

**Summary:**

This manuscript proposes a new backdoor method for two typical scenarios: crowd source annotation and knowledge distillation. The adversaries are assumed to control the training dataset and inject a backdoor through label poisoning. Specifically, the method train several backdoored models using data poisoning methods. Then, it optimize continuous logits to align clean samples and poisoned ones. Finally, it convert the continuous logits to discrete labels. Experimental results on two datasets and two resnet models demonstrate the effectiveness of the proposed mothod.

**Strengths:**

1. The idea of poisoning labels instead of samples is interesting and novel.
2. The manuscript has conducted several exeperiments to evaluate the proposed method.

**Weaknesses:**

1. The two scenarios are common in backdoor learning. Data posioning methods can also be exploited in such scenarios.
2. FLIP needs to train expert models first, where data poisoning models are trained as experts. It would be better to explain why not use data poisoning directly. The necessity of applying FLIP is not convincing.
3. Though FLIP is a new backdoor scheme, from my perspective, it should be compared with data poisoning backdoor models. It's hard to evaluate the performance of FLIP. In other words, FLIP needs to train data poisoning models first, what are the advantages and gains of FLIP compare its first step?
4. Larger models and datasets should be used to evaluate the models.

**Questions:**

1. The advantages of FLIP over previous backdoor methods should be clarified from both motivation and experiment aspects.
2. How much additional computation caused/overhead caused by FLIP compared to a single data poisoning model?
3. Why not conduct experiments on larger datasets and models?

**Limitations:**

1. The authors have not mentioned the limitations of this work. It is worth noting that the benefits of label poisoning backdoor method is not clear. The two motivation scenarios can be also attacked by existing data poisoning methods.
2. There could be potential negative societal impact of this work, if the proposed method is utilized by malicious attackers.

---

> ### Author Rebuttal · Authors · 2023-08-09
>
> We wish to thank the reviewer and address their comments point-by-point below:
>
> We will first clarify our threat model. This will explain the advantage of FLIP and why existing attacks cannot be applied in this setting. In both the crowdsourcing and knowledge distillation scenarios, we assume that:
> 1. The adversary provides the trainer soft or hard labels to (a subset of) the training data.
> 2. The trainer then uses the clean images from the training data and the corrupted labels provided by the adversary to train a model from a random initialization.
>
> Explicitly, the reason the trainer has access to the clean images is that, in both the crowdsourcing and distillation scenarios, the trainer has full control of the images in their training data. In both scenarios, however, the labels are provided by potentially adversarial third parties. This attack surface is different from typical data poisoning or backdoor attacks since the adversary can only corrupt the labels. This is what makes the problem interesting and novel, and this is the reason none of the standard backdoor attacks can be applied. They require corrupting the images, which is not permitted under our threat model. We want to emphasize that FLIP is the first attack that only corrupts labels for a given trigger of choice.
>
> Since there is some variation in terminology surrounding data poisoning and backdoors, we want to establish the convention that data poisoning refers to attacks intended to lower the accuracy on in-distribution data, while backdoor attacks are intended to preserve in-distribution accuracy while controlling the behavior on out-of-distribution data which contain a trigger.
>
> We address each comment below:
> ### Weaknesses:
> 1. While it is true that crowd-sourced data (and distillation with an attacker-provided distillation set) is a common scenario for backdoor attacks, our setting further limits the abilities of the attacker. In our crowd-sourcing setting, the attacker receives a copy of the images to label and provides adversarial labels. Likewise, in our distillation setting the attacker provides a teacher model which is evaluated on the distillation set (and again the attacker cannot modify the distillation images). *Data poisoning attacks*, are common under our threat model where only labels are corrupted, but it is not clear how to create *backdoors* in this setting, where predictions are changed for specific attacker defined triggers in the input. This makes the proposed attack novel and interesting. This differs from the common scenario of backdoor attacks where an adversary injects training examples with corrupted images and corrupted labels.
> 2. We refer to our response above, where we explain why typical *backdoor* attacks cannot be applied since the attacker has no control over the images that the trainer uses in training. FLIP uses *backdoored* models to find which labels to corrupt, and only the corrupted labels are passed onto the model trainer. The backdoored model is never sent to the trainer.
>    1. In the crowd-sourced annotation setting, the attacker provides poisoned labels only.
>    2. In the distillation setting, we find that distillation erases standard backdoors. If we use an expert directly as the teacher model, we achieve a PTA of only 0.02%.
> 3. We refer to our response above and want to emphasize that only the labels are sent to the trainer under our threat model. Hence, existing attacks cannot be applied.
> 4. We refer to the general rebuttal above for additional results using the larger VGG and Vision Transformer architectures as well as the larger Tiny ImageNet dataset.
>
> ### Questions:
> 1. We refer to our response above and want to emphasize that previous backdoor attacks require the attacks to corrupt the images in the training data, which is not allowed under our threat model. FLIP uses existing backdoor attacks (together with trajectory matching) only to find the corrupted labels. Only the labels are sent to the trainer.
> 2. There are two sources of overhead in FLIP: (i) training $m\geq1$ experts and (ii) trajectory matching. For the former, Table 4 in Section 4.5 shows that FLIP exhibits robust performance even when $m=1$. In addition, as we show in **Experiments on $K$** in the main rebuttal, for a successful attack these experts need very few epochs of training.
> Empirically, as described in the supplementary material, for the majority of our experiments, we trained our experts for 20 epochs so that each expert model took no longer than 10 minutes on A40 and 2080ti GPUs to train. Now, the trajectory matching step, on average, took around 25 minutes for the 25 iterations of the algorithm we use. Altogether, this amounts to just over half of a GPU-hour, comparable to the amount of time it took to fully train a model.
> 3. We refer to our response for weakness 4.
>
> ### Limitations:
> 1. We refer to our response above and emphasize that standard backdoor attacks cannot be applied when only label corruption is allowed, as in our threat model. This is the benefit of FLIP over existing attacks. FLIP only corrupts the labels. There are no existing other attacks to compare against. So in Figure 2, we compare against a baseline we came up with, which corrupts labels of images whose inner product with the trigger is large.
>
>    As the reviewer insightfully pointed out, the limitation of FLIP is the additional computational cost in finding the corrupted labels.
> 2. We believe our work will inspire machine learning practitioners to adopt secure measures and motivate further research into backdoor defenses.

---

> > ### Comment · Reviewer_tDVF · 2023-08-14
> > **Thanks for the response**
> >
> > Thanks for the clarification. The repsonse has addressed my concerns. The scenarios of label poisoning are interesting. I would like to change my score to 5. Please clarify the threat model in the paper and add the additional results to the paper.

---

### Author Rebuttal · Authors · 2023-08-09

## Overall Rebuttal: these are additional experiments we ran that were commonly asked by multiple reviewers.
1. **Defenses:** As many of our reviewers judiciously pointed out, FLIP’s resilience to various backdoor defense strategies is of interest. To this end, we evaluate FLIP on CIFAR-10 with all three trigger types on three popular defenses: kmeans [R1], PCA [68], and SPECTRE [27]. We find that SPECTRE is quite effective in mitigating FLIP, whereas the other two defenses fail on the periodic and Turner triggers. We emphasize that even strictly stronger attacks that are allowed to corrupt the images fail against SPECTRE. In any case, we hope that our results will encourage practitioners to adopt strong security measures such as SPECTRE in practice, even under the crowdsourcing and distillation settings with clean images. In our eyes, finding strong backdoor attacks that can bypass SPECTRE is a rewarding future research direction.

   *sinusoidal:*
   ||$150$|$300$|$500$|$1000$|$1500$
   -|-|-|-|-|-
   kmeans|$92.28/10.8$|$92.15/55.6$|$91.68/84.8$|$90.78/96.3$|$90.42/86.3$
   PCA|$92.34/11.7$|$91.95/58.8$|$91.54/85.3$|$90.84/98.4$|$90.40/79.4$
   SPECTRE|$92.50/00.2$|$92.55/00.2$|$92.43/00.2$|$92.06/01.3$|$91.47/01.7$

   *pixel:*
   ||$150$|$300$|$500$|$1000$|$1500$
   -|-|-|-|-|-
   kmeans|$92.13/02.7$|$91.82/04.9$|$91.36/08.3$|$92.37/01.5$|$88.60/30.4$
   PCA|$92.14/02.7$|$91.83/05.2$|$91.73/05.9$|$92.26/02.1$|$92.09/02.5$
   SPECTRE|$92.57/00.0$|$92.42/00.1$|$92.54/00.0$|$92.34/00.1$|$92.27/00.1$

   *Turner:*
   ||$150$|$300$|$500$|$1000$|$1500$
   -|-|-|-|-|-
   kmeans|$92.32/21.2$|$92.06/86.5$|$91.70/95.9$|$90.75/96.0$|$90.01/96.4$
   PCA|$92.25/36.8$|$91.97/95.2$|$91.63/96.8$|$90.79/99.6$|$89.95/98.3$
   SPECTRE|$92.42/00.1$|$92.36/00.1$|$92.17/00.3$|$91.45/00.7$|$90.70/03.4$


2. **Experiments on Larger Models + Transformers.** In response to insightful comments by our reviewers we have added three experiments using two larger architectures: (1) VGG-19 (144M parameters) and (2) Vision Transformer (86M). We present results using discrete labels and the sinusoidal trigger on CIFAR-10.
In the first experiment (Row 1) we show that using only 5 VGG-19 experts, we can backdoor a trainer’s VGG-19 model with high success rate.
For the second experiment (Row 2), we demonstrate that our method is robust to knowledge of the trainer’s architecture (ala Table 3 in the original work). In particular, perhaps surprisingly, using labels generated with ResNet-32s, an attacker can successfully backdoor a user’s VGG-19 model.
In the final experiment (Row 3), we repeat the second, substituting the VGG model for a pretrained Vision Transformer. In addition, instead of training the transformer from scratch we finetune the last few layers with our corrupted labels. We find, again, that our method is successful.
||$150$|$300$|$500$|$1000$|$1500$
-|-|-|-|-|-
r32 $\rightarrow$ vit|$95.42/01.6$|$95.29/06.4$|$95.06/14.5$|$94.67/31.1$|$94.27/40.2$
vgg $\rightarrow$ vgg|$92.78/02.1$|$92.45/07.6$|$92.28/18.9$|$91.53/33.7$|$90.33/47.9$
r32 $\rightarrow$ vgg|$92.76/02.7$|$92.67/10.0$|$92.28/23.1$|$91.41/47.5$|$90.63/63.0$

3. **Experiments on Larger Datasets.** We also added an experiment on Tiny ImageNet (100,000 points). Our results again use discrete labels and the sinusoidal trigger. The attack is again successful.
||$150$|$300$|$500$|$1000$|$1500$
-|-|-|-|-|-
|r18|$61.47/10.6$|$61.23/31.6$|$61.25/56.0$|$61.45/51.8$|$60.94/57.0$

4. **Experiments on $K$.** To aid our discussion, we also conducted an experiment on our hyperparameter $K$ which describes the number of iterations each expert is trained for. As shown below, surprisingly, the attack is still successful when experts are only trained for a single ($K = 1$) iteration. We remark that this is possibly a result of checkpoints later on in training drifting away from the student model training trajectory.
$K$|$150$|$300$|$500$|$1000$|$1500$
-|-|-|-|-|-
$1$|$92.33/17.0$|$92.06/42.8$|$91.59/66.1$|$90.73/85.0$|$89.62/86.8$
$5$|$92.27/12.7$|$92.04/54.1$|$91.66/90.3$|$90.72/98.0$|$89.80/99.6$
$10$|$92.38/09.6$|$92.12/55.7$|$91.65/89.9$|$90.67/99.5$|$89.81/99.8$
$20$|$92.26/12.4$|$92.09/54.9$|$91.73/87.2$|$90.68/99.4$|$89.87/99.8$
$50$|$92.41/08.1$|$92.03/48.0$|$91.72/93.1$|$90.87/99.4$|$90.03/99.8$

[R1] Chen, B., Carvalho, W., Baracaldo, N., Ludwig, H., Edwards, B., Lee, T., Molloy, I., and Srivastava, B., “Detecting backdoor attacks on deep neural networks by activation clustering.”, arXiv preprint arXiv:1811.03728, 2018a.

---

### Decision · Program_Chairs · 2023-09-21

**Decision:**

Accept (poster)

**Comment:**

The paper shows how to create model backdoors when the attacker can only manipulate labels and not the inputs (images) themselves.
This is an interesting and under-explored threat model that nicely matches the way in which many crowdsourced image datasets are created.
The reviewers noted some gaps in the evaluation, e.g., a lack of evaluation of defenses, or few model architectures considered. I encourage the authors to add such results to their paper.